# Patch-as-Decodable-Token: Towards Unified Multi-Modal Vision Tasks in MLLMs

**Yongyi Su**[1,2]  **Haojie Zhang**[1,3]  **Shijie Li**[2*]  **Nanqing Liu**[4]  **Jingyi Liao**[2,5]  **Junyi Pan**[3]
**Yuan Liu**[1]  **Xiaofen Xing**[1]  **Chong Sun**[3]  **Chen Li**[3]  **Nancy F. Chen**[2]  **Shuicheng Yan**[6]
**Xulei Yang**[2]  **Xun Xu**[2*]

[1] South China University of Technology
[2] Institute for Infocomm Research (I$^2$R), A*STAR
[3] WeChat Vision, Tencent Inc.
[4] Yunnan Normal University
[5] Nanyang Technological University
[6] National University of Singapore

## Abstract

Multimodal large language models (MLLMs) have advanced rapidly in recent years. However, existing approaches for vision tasks often rely on indirect representations, such as generating coordinates as text for detection, which limits performance and prevents dense prediction tasks like segmentation. To overcome these challenges, we introduce Patch-as-Decodable Token (PaDT), a unified paradigm that enables MLLMs to directly generate both textual and diverse visual outputs. Central to PaDT are Visual Reference Tokens (VRTs), derived from visual patch embeddings of query images and interleaved seamlessly with LLM's output textual tokens. A lightweight decoder then transforms LLM's outputs into detection, segmentation, and grounding predictions. Unlike prior methods, PaDT processes VRTs independently at each forward pass and dynamically expands the embedding table, thus improving localization and differentiation among similar objects. We further tailor a training strategy for PaDT by randomly selecting VRTs for supervised fine-tuning and introducing a robust per-token cross-entropy loss. Our empirical studies across four visual perception and understanding tasks suggest PaDT consistently achieving state-of-the-art performance, even compared with significantly larger MLLM models. The code is available at
`https://github.com/Gorilla-Lab-SCUT/PaDT`.

## 1 Introduction

Fine-grained image perception and understanding, which aim to associate specific image regions with contextual information, such as semantic or instance, is a fundamental task in computer vision and serves as a cornerstone for numerous applications. Classical vision models (Ren et al., 2015; Redmon et al., 2016; Carion et al., 2020) remain state-of-the-art for pure detection and segmentation tasks, but they lack flexible language interaction and understanding, thus prohibiting open vocabulary oriented visual reasoning tasks. At an earlier stage, inspired by CLIP (Radford et al., 2021), many vision-language detectors such as GLIP (Li et al., 2022b) and Grounding DINO (Ren et al., 2023) incorporate language information to detect arbitrary classes. However, these methods remain vision-centric backbones augmented with language, and thus struggle to handle more complex textual descriptions and are limited to structured output.

Recent advances have led to powerful multi-modal large language models (MLLMs) (Alayrac et al., 2022; Li et al., 2023; Liu et al., 2024c; Bai et al., 2025; Zhu et al., 2025) that couple vision encoders with Large Language Models(LLMs). Pretrained on massive multimodal datasets, these models encode rich prior knowledge and provide a strong foundation for visual perception and understanding, as illustrated in Fig. 1. To conform with the textual output space of LLMs, most existing

---

*Correspondence to <xu_xun@a-star.edu.sg> and <Li_Shijie@a-star.edu.sg>. This work was done during Yongyi Su's visit to I$^2$R and Haojie Zhang's intern in WeChat Vision.

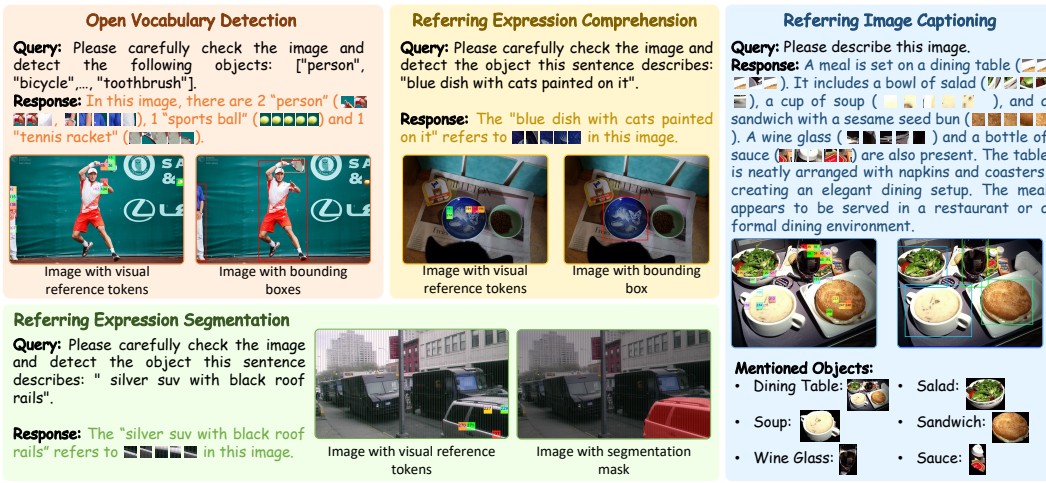

Figure 1: Illustration of unified visual/textual token prediction for MLLM powered visual perception and understanding.

MLLMs (Liu et al., 2025a; Bai et al., 2025; Zhu et al., 2025) serialize detected regions into bounding box coordinates, expressed in textual form, such as $[x_1, y_1, x_2, y_2]$. While straightforward, this strategy introduces several challenges. First, output formats are often inconsistent across samples even under the same prompt, as illustrated in Fig. 2(a), thereby increasing the difficulty of parsing and structured output. Second, numerical coordinate representations provide precise spatial descriptions but lack semantic alignment between textual and visual modalities, as shown in Fig. 2(b). This inherent misalignment can lead to repetition or hallucination between coordinate and actual visual targets (Jiang et al., 2024b). Moreover, since numerical coordinate representations are mapped into discrete textual tokens, a single coordinate value may be split into several unrelated tokens, as shown in Fig. 2(b). These discontinuous coordinate tokens can hinder prediction accuracy, e.g., fragmented numbers.

In this work, we introduce a unified paradigm, **Patch-as-Decodable Token (PaDT)**, which enables MLLMs to directly generate both textual and diverse visual targets in a unified yet flexible way. For this purpose, we propose the **Visual Reference Tokens (VRTs)**, which can be seamlessly interleaved with LLM's output textual tokens. VRTs are generated by the proposed Dynamic Embedding Module, adapted directly from the original visual patch embeddings. In this way, they occur in a feature space consistent with the original LLM, while each VRT explicitly corresponds to a specific image patch within the query image. Thus, VRTs can be naturally interpreted within the LLMs feature space, allowing detected objects to be represented by multiple VRTs in a fine-grained manner. Based on this design, PaDT owns the inherent ability to predict diverse visual outputs, e.g. semantic masks and bounding boxes. Specifically, MLLMs only need to predict a subset of VRTs, which are then decoded into the final structured visual outputs by a lightweight decoder. A prior art (Ma et al., 2025) attempted to empower LLMs to output image patch tokens, discretized by a global codebook, to represent the target within the image. However, this approach remains limited in flexibility and generality due to maintaining a global codebook. First, there is a risk of predicting visual tokens that do not appear in the query image. Moreover, the decoded visual token does not have unique correspondence in the query image, thus risking misalignment between predicted visual tokens and query image tokens, e.g., confusion between similar objects in the image. In contrast, PaDT processes VRTs independently at each forward pass, making it more efficient. By maintaining a high-level feature space aligned with that of LLMs and preserving unique positional information for each image region, PaDT ensures coherent predictions as illustrated in Fig. 2(c). Moreover, as shown in Fig. 2(d), VRT predictions over objects exhibit great spatial continuity.

To enable PaDT to achieve strong performance, we design an effective fine-tuning strategy and propose a robust per-token cross-entropy loss tailored for the proposed visual reference token, which stabilizes training and mitigates overfitting. Notably, our 3B model surpasses the previous state-of-the-art by 19.0 mAP on COCO detection and achieves an average accuracy of 93.6 on the referring expression comprehension (REC) task, outperforming the much larger 78B InternVL3 model.

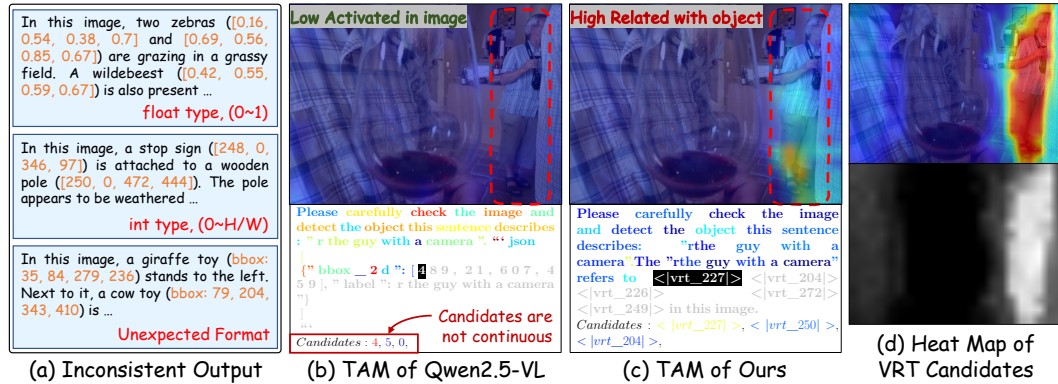

Figure 2: (a) Previous methods yield inconsistent output formats due to free-form box representations even under the same prompt. (b) Token Activation Map (TAM) (Li et al., 2025) reveals less semantic relationship between textual box representations and textual/visual information, while converting continuous numbers into discrete tokens further introduces discontinuities. (c) With PaDT denoting objects with VRTs, semantic alignment is preserved and the output becomes more unified and natural. (d) The heatmap of <VRT_227> further demonstrates continuous and object-consistent predictions within the input image.

The main contributions of this work can be summarized as follows:

- We introduce a unified paradigm, Patch-as-Decodable Token (PaDT), which enables MLLMs to directly generate both textual and diverse visual targets in a unified yet flexible way. With the proposed Visual Reference Token (VRT), our method achieves superior performance across diverse fine-grained image perception and understanding.

- We propose a lightweight yet robust VRT-based decoder, termed the PaDT Decoder. Given the generated VRTs, it can uniformly decode diverse fine-grained structured visual outputs, such as segmentation masks and bounding boxes.

- We propose an effective fine-tuning strategy together with a robust per-token cross-entropy loss. PaDT achieves the state-of-the-art performance on a wide range of visual perception and understanding tasks. The effectiveness is validated beyond perception tasks but also a customized image captioning task.

## 2 RELATED WORK

**Multimodal Large Language Models.** With the rapid development of large language models, multimodal LLMs (MLLMs) have emerged as powerful systems for vision-language reasoning (Alayrac et al., 2022; Achiam et al., 2023; Liu et al., 2023; Zhu et al., 2023; Zhang et al., 2024a; Lian et al., 2025; Bai et al., 2025). Early milestones such as CLIP (Radford et al., 2021) and ALIGN (Jia et al., 2021) demonstrated the effectiveness of large-scale contrastive pretraining for joint vision-text representations. BLIP-2 (Li et al., 2022a) further improved alignment through the Q-former design. More recently, instruction-tuned MLLMs including LLaVA (Liu et al., 2023) and MiniGPT-4 (Zhu et al., 2023) leverage multimodal instruction data, yielding strong performance in open-ended visual question answering and reasoning. Building on these foundations, subsequent works extend capabilities to higher-resolution image understanding (e.g., LLaVA-Next (Liu et al., 2024c), LLaVA-UHD (Guo et al., 2024)), diverse instruction sets (Ye et al., 2023), multi-image (Jiang et al., 2024a; Li et al., 2024) and video inputs (Lin et al., 2023a; Chen et al., 2024a), as well as new pretraining objectives and architectural designs (Fang et al., 2023; Wang et al., 2023b). Collectively, these advances establish MLLMs as versatile general-purpose models for multimodal reasoning.

**MLLMs for Visual Perception & Understanding.** Despite their broad capabilities, general-purpose MLLMs remain limited in fine-grained perception tasks. This stems largely from vision encoders reliance on fixed patch grids (Dehghani et al., 2023; Fang et al., 2023; Wang et al., 2023b), which often blur local details and impair tasks such as object localization, counting, or OCR. To mitigate this, adaptive tiling strategies, such as NaViT-style patch dropping and AnyRes (Luo et al.,

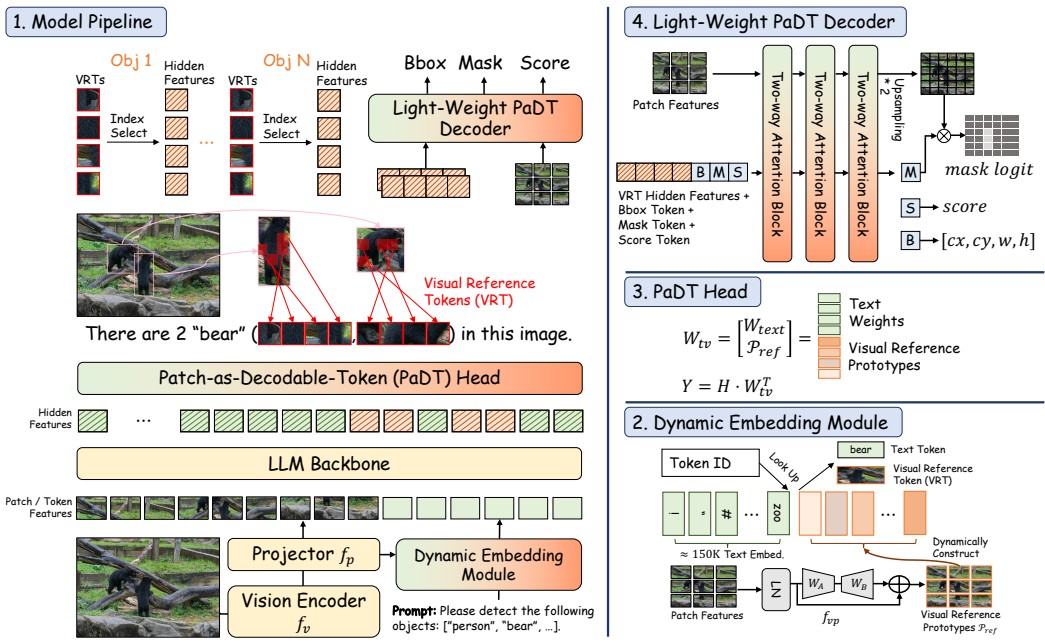

Figure 3: The framework of PaDT model.

2023; Chen et al., 2024b; Liu et al., 2024a), allow flexible handling of variable-resolution image tiles, leading to improved spatial resolution. Another line of work explores reinforcement learning to enhance perception and reasoning, exemplified by VLM-R1 (Shen et al., 2025), Visual-RFT (Liu et al., 2025b), VisRL (Chen et al., 2025), and Seg-R1 (You & Wu, 2025). These approaches achieve better generalization and emergent capabilities such as segmentation and grounding. Prior works have primarily relied on reinforcement learning (Chen et al., 2025) or instruction tuning (Jiang et al., 2024b) to strengthen visual reasoning, yet the potential of leveraging learned queries as anchors for visual perception remains underexplored. Moreover, designing a unified architecture that seamlessly accommodates diverse vision tasks continues to be an open challenge.

**Unified Visual Tokenization.** A complementary research direction focuses on unifying visual and linguistic representations through multi-granular tokenization. At the region level, methods convert object boxes or masks into geometric tokens (Chen et al., 2023b; Xuan et al., 2024; Peng et al., 2023; You et al., 2023) or learnable proxies (Zhang et al., 2024b; Yuan et al., 2024; Chen et al., 2023a; Rasheed et al., 2024), often grounded by detectors or SAM (Kirillov et al., 2023), thereby enabling more precise vision-language grounding. At the patch level, models such as the Emu series (Sun et al., 2023) and LaVIT (Jin et al., 2024) treat CLIP-derived patch features as visual vocabularies for denser alignment. Recent works further introduce autoregressive quantization of image patches (Team, 2024; Sun et al., 2024), discretizing pixels into visual sentences to support efficient cross-modal modeling, with even finer-grained tokenization explored in (Ma et al., 2025). While these approaches approximate linguistic structures via region, instance, or pixel tokens, deeper semantic integration between vision and language is still limited. To address this, we propose a dynamic multimodal token space that enables close correspondence between language tokens and visual patches under a unified autoregressive modeling paradigm.

## 3 METHODOLOGY

### 3.1 REVISITING MULTIMODAL LARGE LANGUAGE MODELS

A Multimodal Large Language Model (MLLM) augments a Large Language Model (LLM) with a visual encoder, enabling it to perform not only general-purpose reasoning but also visual perception (Alayrac et al., 2022; Liu et al., 2024c; Bai et al., 2025). Given an image $I \in \mathbb{R}^{H \times W \times 3}$ and a text sequence $\mathbf{T} = (t_1, \ldots, t_m)$, the MLLM autoregressively generates an output sequence

$\mathbf{Y} = (y_1, \ldots, y_t)$. An image encoder $f_v$, typically a Vision Transformer (ViT) (Dosovitskiy et al., 2020), partitions $\mathbf{I}$ into $N$ non-overlapping patches $\{P_n\}_{n=1}^N$, which are subsequently encoded into embeddings $F_v = f_v(I) \in \mathbb{R}^{N \times d_v}$. A projector $f_p$ then aligns dimensions and downsamples, yielding $F_{patch} = f_p(F_v) \in \mathbb{R}^{N' \times d}$. For instance, Qwen2.5-VL adopts nearest-neighbor patch merging in the 2D patch space, resulting in $N' = \frac{1}{4}N$. The image embeddings are then fused with the text embeddings $E_{text}(\mathbf{T}) \in \mathbb{R}^{m \times d}$ to form a hybrid textual-visual representation $Z = [F_{patch}; E_{text}(\mathbf{T})]$. Here, $E_{text} \in \mathbb{R}^{V_{text} \times d}$ denotes the text embedding table that maps each text token to its corresponding feature vector. The resulting multimodal representation $Z$ is subsequently fed into a transformer-based LLM (Alayrac et al., 2022; Liu et al., 2024c; Bai et al., 2025). At timestep $t$, the hidden state $h_t$ produces the next-token distribution:

$$p(y_t|I, \mathbf{T}, y_{<t}) = \text{softmax}(W_{text} \cdot h_t), \tag{1}$$

with $W_{text} \in \mathbb{R}^{V_{text} \times d}$ denoting classifier weights.

**Limitations of Text-based Vision Prediction.** Current MLLMs are restricted to accepting textualvisual representations as input and producing only textual outputs, owing to their compatibility with the underlying LLM architecture. This limitation is suboptimal for structured vision tasks such as object detection and image segmentation. Specifically, current MLLMs (e.g., Qwen2.5-VL (Bai et al., 2025), InternVL3 (Zhu et al., 2025)) serialize visual targets into strings at output side. This leads to two major issues. First, outputs vary in format (absolute vs. normalized coordinates, JSON-style vs. free-form), complicating parsing and structured output, as shown in Fig. 2(a). Second, numerical coordinate representations are mapped into discrete textual tokens which are generated digit by digit (e.g., "489" "4, 8, 9"). This disrupts numerical continuity and may hinder prediction accuracy (Fig. 2(b)). More importantly, while this numerical representation effectively describes spatial information precisely, it lacks semantic information, which is crucial for image understanding tasks. This inherent mismatch, revealed through token activation analyses (Li et al., 2025) as illustrated in Fig. 2(b), can lead to errors such as repetition or hallucination in dense prediction tasks (Jiang et al., 2024b).

### 3.2 Visual Reference Token

We propose the **Patch-as-Decodable-Token** (PaDT) framework, which introduces **Visual Reference Tokens** (VRTs), a unified tokenization scheme that embeds visual patches directly as decodable tokens within the autoregressive generation process. PaDT extends conventional MLLMs with three key components: (1) *Dynamic Embedding Module* augments the textual vocabulary codebook with visual patches, specific VRTs, at each forward pass, yielding a multi-modal codebook. (2) With this multi-modal codebook and the proposed *PaDT Head*, VRTs become both embeddable at the input side and decodable at the output side, resulting in a unified and natural format. (3) A lightweight *PaDT Decoder* is proposed to convert variable VRTs into diverse visual representations, such as bounding boxes and masks, enabling downstream tasks including detection, segmentation, and grounding. This further enhances both the robustness and flexibility of the proposed method.

#### 3.2.1 Unified Multi-modal format with VRTs

A core challenge is to ensure that VRTs can by interpretable by LLMs, being both *embeddable* in the input space and *decodable* in the output space. Prior work, e.g., ClawMachine (Ma et al., 2025) relies on pretrained discrete visual tokenizers (Jin et al., 2024). It inserts the entire codebook, which contains a massive number of tokens, into the LLM embedding table and forces the LLM to map its high-level semantic feature space to tokens representing low-level image patches. Thus, this method is limited by (i) a fixed dataset-level codebook expansion which contains massive tokens that ignore patch-specific cues such as spatial location, and (ii) ambiguity arising from the lack of high-level semantics when visually similar patches from different objects maybe mapped to the same token.

**Dynamic Multi-Modal Codebook Expansion.** To avoid the above limitations, rather than introducing a standalone codebook, we reuse the extracted visual tokens from the input image, which already preserve rich semantic information. Since each visual token explicitly corresponds to an image patch, at each forward pass only the tokens from the current query image are dynamically expanded into the original textual codebook, instead of memorizing all possible visual patterns through a fixed codebook. Specifically, in the proposed *Dynamic Embedding Module*, original patch features

$F_{patch} \in \mathbb{R}^{N' \times d}$ are projected by a lightweight module $f_{vp}$ into visual reference prototypes $\mathcal{P}_{ref}$. $f_{vp}$ consists of a LayerNorm and a low-rank linear projection. These prototypes are then concatenated with text embeddings to form a dynamic embedding table as,

$$E_{dyn} = [E_{text}; \mathcal{P}_{ref}], \quad \mathcal{P}_{ref} = f_{vp}(F_{patch}) \in \mathbb{R}^{N' \times d}. \tag{2}$$

**Unified Input and Output Format.** With the above Multi-Modal Codebook, both textual and visual information can be input and output in a unified way. On the input side, query image tokens are indexed in the Multi-Modal Codebook and converted into the corresponding VRTs, which are then embedded into the textual input to the LLM. Since VRTs are adapted from the original image tokens, they share a feature space that is similar to the LLMs representation space, which simplifies training compared to ClawMachine (Ma et al., 2025). On the output side, to enable the original textual classifier to output expanded indices, the *PaDT Head* is proposed to augment the classifier with $\mathcal{P}_{ref}$, yielding

$$W_{tv} = [W_{text}; \mathcal{P}_{ref}] \in \mathbb{R}^{(V_{text} + N') \times d}. \tag{3}$$

This joint design allows VRTs to be embedded as inputs and decoded as outputs, enabling the model to insert patch-level references directly into the autoregressive sequence. Building on this, we propose a robust strategy that represents detected objects with several (but not all) VRTs placed on them, and then decodes fine-grained representations such as bounding boxes or masks through the lightweight PaDT Decoder introduced below. This strategy is shown to be more robust and effective in our experiments. Template examples for each vision task are provided in Appendix A.2.

### 3.2.2 LIGHT-WEIGHT PaDT DECODER

Considering that only several VRTs on a detected object are predicted, a visual decoder is needed to convert these predicted VRTs into task-specific outputs. For this purpose, we introduce a lightweight vision task decoder, implemented as a stack of three two-way attention blocks (Fig. 4(b)). The decoder takes as input the hidden features of predicted VRTs from the final LLM layer. These features are grouped into object queries, where each group corresponds to a sequence of VRTs separated by intervening text tokens (Fig. 4(a)). To enable task-specific decoding, we inject three learnable tokens, *bounding box*, *mask*, and *score* tokens, into each group of object queries. After passing through the three attention blocks, each task token is projected into its respective output space, producing bounding boxes, segmentation masks, and confidence scores.

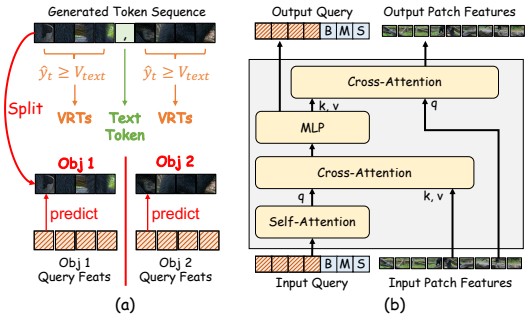

Figure 4: Illustration for PaDT decoder.

### 3.2.3 TRAINING STRATEGY

**Robust Per-token Cross-Entropy Loss**. For the autoregressive output of the MLLM, we adopt the standard supervised fine-tuning paradigm with a per-token cross-entropy loss:

$$\mathcal{L}_{CE} = \frac{1}{T} \sum_t -\log p(\hat{y}_t \mid I, \mathbf{T}, y_{<t}) = -\log \mathrm{softmax}_{GT}(W_{tv} \cdot h_t), \tag{4}$$

where $\hat{y}_t$ denotes the ground-truth token at step $t$, $h_t$ is the hidden state, and $W_{tv}$ projects to the token vocabulary.

Unlike prior work that uses all foreground visual tokens as supervision (Ma et al., 2025), we propose to randomly sample $N_{vrt}$ foreground tokens for each forward pass. This sampling strategy increases the diversity of supervision and prevents the model from overfitting to a fixed set of tokens, thereby improving generalization. To implement this, we introduce a foreground mask $M \in \{0, 1\}^{T \times N'}$, where $M_{t,n} = 1$ indicates that token $n$ at step $t$ was not selected. For such tokens, we suppress their contribution to the loss by masking their logits:

$$l'_t = W_{tv} \cdot h_t, \quad l'_{t,n+V_{text}} = -\infty \text{ if } M_{t,n} = 1. \tag{5}$$

This effectively removes the masked tokens from the softmax normalization, ensuring they are neither rewarded nor penalized. The resulting robust cross-entropy loss is:

$$\mathcal{L}_{CE}^{robust} = -\log \text{softmax}_{GT}(l'_t). \tag{6}$$

By combining random sampling with masked supervision, this objective improves robustness and encourages the model to explore diverse valid visual references during training.

**Task-specific Losses**. For structured outputs from vision task decoder, we adopt task-specific objectives i.e. $\mathcal{L}_{bbox}$, $\mathcal{L}_{mask}$ and $\mathcal{L}_{score}$ following (Kamath et al., 2021; Kirillov et al., 2023). More implemented details about the task-specific losses are given in the Appendix A.4. The final training objective of PaDT is

$$\mathcal{L} = \mathcal{L}_{CE}^{robust} + \mathcal{L}_{bbox} + \mathcal{L}_{mask} + \mathcal{L}_{score}. \tag{7}$$

# 4 EXPERIMENT

**Tasks and Datasets.** We evaluate PaDT across a diverse set of visual perception & understanding tasks. Specifically, we consider: (i) referring expression comprehension and referring expression segmentation on RefCOCO, RefCOCO+, and RefCOCOg (Mao et al., 2016; Yu et al., 2016); (ii) open-vocabulary detection on COCO 2017 (Lin et al., 2014); and (iii) referring image captioning (RIC), for which we construct a new benchmark by re-annotating COCO with visionlanguage model (VLM) supervision. Further dataset details are provided in Appendix A.1.

**Architecture and Training Details.** We adopt Qwen2.5-VL (Bai et al., 2025) as the base model and conduct experiments with both 3B and 7B variants to evaluate scalability. Based on existing dataset annotations, at each training step we randomly sample $N_{\text{vrt}} = 5$ visual reference tokens from the foreground mask of each target to construct the ground-truth MLLM sequence. If segmentation masks are unavailable, VRTs are instead sampled within the bounding box. The ground-truth token templates are provided in Appendix A.2. Training is performed on a single node with eight 96GB GPUs, using a batch size of 16 per GPU. We set the learning rate to $2 \times 10^{-5}$ and apply gradient checkpointing together with `bfloat16` mixed precision for memory efficiency. FlashAttention-2 (Dao, 2023) is further employed to accelerate attention computation.

Table 1: Results of referring expression comprehension task on RefCOCO/+/g datasets.

| Model Name | Param. | RefCOCO | | | RefCOCO+ | | | RefCOCOg | | Overall |
| --- | --- | --- | --- | --- | --- | --- | --- | --- | --- | --- |
| | | val | test-A | test-B | val | test-A | test-B | val | test | |
| Grounding-DINO-L (Liu et al., 2024d) | - | 90.6 | 93.2 | 88.2 | 82.8 | 89.0 | 75.9 | 86.1 | 87.0 | 86.6 |
| UNINEXT-H (Lin et al., 2023b) | - | 92.6 | 94.3 | 91.5 | 85.2 | 89.6 | 79.8 | 88.7 | 89.4 | 88.9 |
| ONE-PEACE (Wang et al., 2023a) | - | 92.6 | 94.2 | 89.3 | 88.8 | 92.2 | 83.2 | 89.2 | 89.3 | 89.9 |
| InternVL3 (Zhu et al., 2025) | 1B | 85.8 | 90.1 | 81.7 | 76.6 | 84.1 | 69.2 | 82.8 | 82.6 | 81.6 |
| InternVL3 (Zhu et al., 2025) | 2B | 89.8 | 92.6 | 86.4 | 84.0 | 89.2 | 76.5 | 87.6 | 87.2 | 86.7 |
| Qwen2.5-VL (Bai et al., 2025) | 3B | 89.1 | 91.7 | 84.0 | 82.4 | 88.0 | 74.1 | 85.2 | 85.7 | 85.0 |
| Qwen2.5-VL (SFT, (Shen et al., 2025)) | 3B | 88.7 | - | - | 82.3 | - | - | 86.0 | - | - |
| VLM-R1 (Shen et al., 2025) | 3B | 90.1 | 92.3 | 85.2 | 84.2 | 89.4 | 76.8 | 85.6 | 86.8 | 86.3 |
| **PaDT (Ours)** | 3B | 93.2 | 95.3 | 90.1 | 88.5 | 92.4 | 83.5 | 88.2 | 88.5 | 90.0 |
| **PaDT Pro (Ours)** | 3B | **96.0** | **95.5** | **95.0** | **91.8** | **94.8** | **88.4** | **93.6** | **94.0** | **93.6** |
| Shikra (Chen et al., 2023b) | 7B | 87.0 | 90.6 | 80.2 | 81.6 | 87.4 | 72.1 | 82.3 | 82.2 | 82.9 |
| Ferret (You et al., 2023) | 7B | 87.5 | 91.4 | 82.5 | 80.8 | 87.4 | 73.1 | 83.9 | 84.8 | 83.9 |
| Ferret-v2 (Zhang et al., 2024a) | 7B | 92.8 | 94.7 | 88.7 | 87.4 | 92.8 | 79.4 | 89.4 | 89.3 | 89.3 |
| TextHawk2 (Yu et al., 2024) | 7B | 91.9 | 93.0 | 87.6 | 86.2 | 90.0 | 80.4 | 88.2 | 88.1 | 88.2 |
| ClawMachineX (Ma et al., 2025) | 7B | 89.7 | 92.5 | 86.9 | 84.4 | 88.9 | 78.0 | 86.7 | 87.1 | 86.8 |
| Qwen2.5-VL (Bai et al., 2025) | 7B | 90.0 | 92.5 | 85.4 | 94.2 | 89.1 | 76.9 | 87.2 | 87.2 | 86.6 |
| InternVL3 (Zhu et al., 2025) | 8B | 92.5 | 94.6 | 88.0 | 88.2 | 92.5 | 81.8 | 89.6 | 90.0 | 89.6 |
| **PaDT (Ours)** | 7B | 93.1 | 97.2 | 90.4 | 88.8 | 92.8 | 83.2 | 88.2 | 88.8 | 90.1 |
| **PaDT Pro (Ours)** | 7B | **96.6** | **97.4** | **95.6** | **92.8** | **95.2** | **89.4** | **94.6** | **94.2** | **94.5** |
| Ferret (You et al., 2023) | 13B | 89.5 | 92.4 | 84.4 | 82.8 | 88.1 | 75.2 | 85.8 | 86.3 | 85.6 |
| Ferret-v2 (Zhang et al., 2024a) | 13B | 92.6 | 95.0 | 88.9 | 87.4 | 92.1 | 81.4 | 89.4 | 90.0 | 89.6 |
| InternVL3 (Zhu et al., 2025) | 14B | 92.0 | 94.4 | 87.8 | 87.4 | 92.1 | 81.5 | 88.6 | 89.3 | 89.1 |
| CogVLM-Grounding (Wang et al., 2024) | 17B | 92.8 | 94.8 | 89.0 | 88.7 | 92.9 | 83.4 | 89.8 | 90.8 | 90.3 |
| InternVL3 (Zhu et al., 2025) | 78B | 93.4 | 95.4 | 90.3 | 90.1 | 93.8 | 85.3 | 91.5 | 91.5 | 91.4 |

**Multi-Task Scalability.** Joint training across tasks consistently improves performance, indicating strong cross-task generalization. To evaluate multi-task performance and analyze how performance scales with the number of tasks, we train PaDT jointly across all benchmarks, i.e., RefCOCO/+/g, COCO, and RIC, resulting in an enhanced multi-task variant denoted as **PaDT Pro**. Unlike task-specific PaDT models, PaDT Pro can seamlessly switch between tasks by simply altering the prompt.

## 4.1 VISUAL PERCEPTION & UNDERSTANDING TASKS

**Referring Expression Comprehension.** The Referring Expression Comprehension (REC) task evaluates an MLLMs ability to localize objects given natural language descriptions, where a prediction is considered correct if its IoU with the ground-truth box exceeds $50\%$. As shown in Tab. 1, PaDT and PaDT Pro achieve state-of-the-art performance at both 3B and 7B scales. In particular, PaDT Pro (3B) obtains 96.0/95.5/95.0 on RefCOCO, 91.8/94.8/88.4 on RefCOCO+, and 93.6/94.0 on RefCOCOg, surpassing all previous MLLM methods. The overall average of PaDT Pro (3B) reaches 93.6, which is further boosted to 94.5 with the 7B model. Remarkably, both PaDT and PaDT Pro (3B) already outperform the much larger 78B InternVL3 model. These results demonstrate the effectiveness of the visual reference token paradigm, which substantially aligns textual semantics with image patches and thereby improves the precision of object localization in MLLMs.

**Referring Expression Segmentation.** Similar to REC, the Referring Expression Segmentation (RES) task evaluates an MLLMs ability to segment the target object mask given a natural language description. We adopt cIoU as the evaluation metric, and results are reported in Tab. 2. Both PaDT and PaDT Pro achieve the best performance compared with existing methods, even against approaches such as Seg-R1 and Text4Seg+SAM that leverage the powerful SAM segmentation model. With the lightweight PaDT decoder that translates unified visual reference tokens into segmenta-

Table 2: Results of referring expression segmentation task on RefCOCO/+/g datasets.

| Model Name | Param. | RefCOCO | | | RefCOCO+ | | | RefCOCOg | | Overall |
|---|---|---|---|---|---|---|---|---|---|---|
| | | val | testA | testB | val | testA | testB | val | test | |
| X-Decoder (Zou et al., 2023a) | - | - | - | - | - | - | - | 64.6 | - | - |
| SEEM (Zou et al., 2023b) | - | - | - | - | - | - | - | 65.7 | - | - |
| Seg-R1 (You & Wu, 2025) | 3B | 69.9 | 76.0 | 64.9 | 59.1 | 66.8 | 50.9 | 67.3 | 67.9 | 65.4 |
| **PaDT (Ours)** | 3B | 76.1 | 77.4 | 74.7 | 72.7 | 75.1 | 69.3 | 70.5 | 71.1 | 73.4 |
| **PaDT Pro (Ours)** | 3B | 81.3 | 81.5 | 82.2 | 77.6 | 79.4 | 76.3 | 78.1 | 78.5 | 79.4 |
| LAVT (Ye et al., 2023) | 7B | 72.7 | 75.8 | 68.8 | 62.1 | 68.4 | 55.1 | 65.0 | 66.0 | 66.7 |
| LISA (Lai et al., 2024) | 7B | 74.1 | 76.5 | 71.1 | 62.4 | 67.5 | 56.5 | 66.4 | 68.5 | 67.9 |
| PixelLM (Ren et al., 2024) | 7B | 73.0 | 76.5 | 68.2 | 66.3 | 71.7 | 58.3 | 69.3 | 70.5 | 69.2 |
| OMG-LLaVA (Zhang et al., 2024c) | 7B | 75.6 | 77.7 | 71.2 | 65.6 | 69.7 | 58.9 | 70.7 | 70.2 | 70.0 |
| Seg-R1 (You & Wu, 2025) | 7B | 74.3 | 78.7 | 67.6 | 62.6 | 70.9 | 57.9 | 71.0 | 71.4 | 69.3 |
| Text4Seg + CRF (Lan et al., 2025) | 7B | 71.3 | 73.7 | 69.6 | 65.9 | 70.4 | 61.9 | 69.3 | 69.3 | 68.9 |
| Text4Seg + SAM (Lan et al., 2025) | 7B | 78.0 | 80.9 | 74.6 | 71.6 | 77.3 | 66.0 | 74.8 | 74.7 | 74.7 |
| **PaDT (Ours)** | 7B | 78.5 | 79.8 | 77.3 | 75.0 | 77.7 | 71.3 | 73.0 | 73.9 | 75.8 |
| **PaDT Pro (Ours)** | 7B | 86.0 | 86.1 | 86.4 | 82.5 | 84.1 | 80.7 | 83.5 | 83.3 | 84.1 |

Table 3: Results of open-vocabulary detection task on the whole COCO2017 validation set.

| Model Name | Param. | AP@[50:95] | AP@50 | AP@75 | AR@[50:95] | AR@50 | AR@75 |
|---|---|---|---|---|---|---|---|
| InternVL3 (Zhu et al., 2025) | 2B | 6.9 | 11.2 | 7.0 | 14.9 | 20.8 | 15.6 |
| Qwen2.5-VL (Bai et al., 2025) | 3B | 13.7 | 22.1 | 14.2 | 21.8 | 30.5 | 23.3 |
| Qwen2.5-VL-SFT (Shen et al., 2025) | 3B | 17.1 | 27.5 | 17.3 | 25.4 | 35.6 | 26.4 |
| VLM-R1 (Shen et al., 2025) | 3B | 19.2 | 33.1 | 19.0 | 32.2 | 46.9 | 33.6 |
| **PaDT (Ours)** | 3B | 34.0 | 51.2 | 35.8 | 38.5 | 56.1 | 40.4 |
| **PaDT Pro (Ours)** | 3B | 38.2 | 54.9 | 40.5 | 43.9 | 60.6 | 46.4 |
| Qwen2.5-VL (Bai et al., 2025) | 7B | 18.2 | 30.4 | 17.9 | 28.1 | 40.3 | 29.3 |
| LLaVa-NeXT (Liu et al., 2024b) | 7B | 0.7 | 2.2 | 0.3 | 1.3 | 3.3 | 0.8 |
| LLaVa-OneVision (Li et al., 2024) | 7B | 2.2 | 5.8 | 1.1 | 4.1 | 8.8 | 3.2 |
| InternVL3 (Zhu et al., 2025) | 8B | 17.5 | 26.6 | 18.2 | 28.0 | 37.3 | 29.7 |
| **PaDT (Ours)** | 7B | 36.5 | 53.8 | 38.4 | 41.5 | 59.2 | 43.6 |
| **PaDT Pro (Ours)** | 7B | 39.0 | 56.2 | 41.5 | 44.8 | 61.8 | 47.6 |

Table 4: Results of referring image captioning task on RIC validation set.

| Model Name | Param. | Text Metrics | | | | Detection Metrics | |
|---|---|---|---|---|---|---|---|
| | | CIDEr-D | Meteor | ROUGE-L | BLEU-4 | GP | GR |
| LLaVa-OneVision (Li et al., 2024) | 0.5B | 0.058 | 0.088 | 0.185 | 0.052 | 5.2 | 0.5 |
| InternVL3 (Zhu et al., 2025) | 2B | 0.315 | 0.230 | 0.374 | 0.284 | 42.4 | 18.2 |
| Qwen2.5-VL (Bai et al., 2025) | 3B | 0.386 | 0.241 | 0.369 | 0.261 | 61.8 | 6.2 |
| **PaDT (Ours)** | 3B | **1.450** | **0.304** | **0.501** | **0.467** | 81.6 | **45.4** |
| **PaDT Pro (Ours)** | 3B | 1.412 | 0.300 | 0.495 | 0.458 | **82.3** | 45.1 |
| LLaVa-NeXT (Liu et al., 2024b) | 7B | 0.262 | 0.200 | 0.335 | 0.178 | 54.3 | 10.6 |
| LLaVa-OneVision (Li et al., 2024) | 7B | 0.172 | 0.207 | 0.330 | 0.182 | 32.5 | 10.2 |
| Qwen2.5-VL (Bai et al., 2025) | 7B | 0.266 | 0.251 | 0.369 | 0.257 | 60.8 | 19.8 |
| InternVL3 (Zhu et al., 2025) | 8B | 0.208 | 0.207 | 0.373 | 0.249 | 56.6 | 32.1 |
| LLaVa-NeXT (Liu et al., 2024b) | 13B | 0.283 | 0.212 | 0.347 | 0.172 | 55.7 | 6.2 |
| **PaDT (Ours)** | 7B | **1.445** | **0.304** | **0.500** | **0.466** | 77.0 | 45.2 |
| **PaDT Pro (Ours)** | 7B | 1.387 | 0.299 | 0.491 | 0.449 | **82.3** | **45.8** |

Table 5: The ablation study of the proposed components in PaDT.

| Visual Reference Token | | | Training Strategy | | REC | RES |
|---|---|---|---|---|---|---|
| using VRTs | $f_{vp}$ | Task Decoder | $\mathcal{L}_{CE}^{robust}$ | VRTs Selection | RefCOCO val | RefCOCO val |
| – | – | – | – | – | 88.7 | – |
| ✓ | – | PaDT Decoder | ✓ | ✓ | 91.1 | 72.1 |
| ✓ | ✓ | PaDT Decoder | – | ✓ | 92.0 | 75.2 |
| ✓ | ✓ | PaDT Decoder | – | All VRTs | 76.5 | 69.5 |
| ✓ | ✓ | PaDT Decoder | ✓ | All VRTs | 49.1 | 19.8 |
| ✓ | ✓ | PaDT Decoder | ✓ | ✓ | **93.2** | **76.1** |

tion masks, our models consistently outperform prior baselines. Additional qualitative examples are provided in the Appendix A.8.

**Open-vocabulary Detection.** This is a fundamental visual perception task that evaluates an MLLMs ability to perform semantic grounding. As shown in Table 3, most existing MLLMs struggle with this task, showing low precision and recall. For instance, Qwen2.5-VL (3B) achieves only 13.7 mAP, and InternVL3 (8B) reaches 17.5 mAP on the COCO2017 validation set. Our PaDT and PaDT Pro substantially advance the state of the art. PaDT Pro (3B) achieves 38.2 mAP, while the 7B variant further improves to 39.0 mAP, nearly doubling the performance of prior best methods. These gains highlight the effectiveness of visual reference tokens in strengthening semantic association and object localization.

**Referring Image Captioning.** To validate both the visual understanding and grounding ability, we conduct experiments on our RIC dataset. As shown in Table 4, PaDT and PaDT Pro (3B) deliver strong improvements, reaching 1.45 CIDEr, 0.304 Meteor, 0.501 ROUGE-L, 0.467 BLEU-4, and top detection scores of 82.3% Greedy Precision (GP) and 45.1% Greedy Recall (GR). The 7B models further extend performance, with PaDT Pro (7B) maintaining competitive caption quality, i.e. 1.39 CIDEr, while achieving the best detection-oriented scores, i.e. 82.3% GP, 45.8% GR. These results suggest that PaDT generates not only fluent captions, but also semantically precise ones grounded in localized visual content.

## 4.2 ABLATION EXPERIMENTS

**Ablation study of Proposed Components in PaDT.** We conduct detailed ablation studies in Tab. 5 using the 3B model with the following observations. i) The first row without VRTs corresponds to supervised fine-tuning on Qwen2.5-VL, directly predicting bounding box coordinates. By integrating VRTs with robust CE loss and random VRTs selection, we observe noticeable improvement in REC (detection task) and RES (segmentation task) being enabled. ii) We further notice that both projection module $f_{vp}$ and robust CE loss are necessary for achieving improved performance. iii) Alternative choice of including all foreground VRTs during training may even harm the performance, probably due to bias towards high density regions.

Figure 5: The illustrations of the mask generations.

Table 6: Performance of using SAM2-L as mask refiner with 3 types of prompts.

| point | box | mask | RefCOCOg val |
|:---:|:---:|:---:|:---:|
| – | – | – | 70.5 |
| ✓ | – | – | 69.9 |
| – | ✓ | – | 74.1 |
| – | – | ✓ | 74.0 |
| ✓ | ✓ | – | 74.9 |
| ✓ | ✓ | ✓ | 76.3 |

Table 7: The generalization analysis and finetuning result of PaDT on COCO2017 validation set.

| Model Name | Objects365 | COCO2017 | AP@[50:95] | AP@50 | AP@75 | AR@[50:95] | AR@50 | AR@75 |
|---|:---:|:---:|:---:|:---:|:---:|:---:|:---:|:---:|
| Qwen2.5-VL | – | – | 13.7 | 22.1 | 14.2 | 21.8 | 30.5 | 23.3 |
| **PaDT** (Zero Shot) | ✓ | – | 16.9 | 23.7 | 18.0 | 21.5 | 30.6 | 22.7 |
| **PaDT** (Task Specific) | – | ✓ | 34.0 | 51.2 | 35.8 | 38.5 | 56.1 | 40.4 |
| **PaDT** (FineTuned) | ✓ | ✓ | **36.5** | **52.2** | **38.8** | **41.3** | **57.4** | **43.6** |

**Effectiveness of Mask Refinement with SAM2-L.** We further analyze the compability of PaDT output with segmentation foundation model, SAM2-L under three schemes. i) Given the VRTs generated by PaDT, we extract their coordinates as point prompts to SAM2-L, denoted as **point**. ii) Using the bounding box and mask generated by PaDT, respectively, as prompt for SAM2-L. We explored different combinations with results in Tab. 6. First, we observe that using point prompt fails to improve upon PaDT, due to the sparse prior information. However, both box and mask prompts are conducive to further improving the results under the help of SAM. Combining multiple prompts yields more significant improvement. Visualizations in Fig. 5 corroborate these findings. The results suggest the segmentation performance can be further enhanced with expert foundation model at the expense of additional inference cost.

**Effectiveness of Pretraining and Task-specific Finetuning.** To evaluate the generalization and data-scaling properties of the PaDT framework, we pretrain on Objects365 (Shao et al., 2019) and subsequently finetune on the COCO dataset. As shown in Tab. 7, PaDT exhibits stronger zero-shot performance than the Qwen2.5-VL base model, and its finetuned version consistently outperforms direct training on task-specific data.

## 5 CONCLUSION

In this work, we proposed Patch-as-Decodable Token (PaDT), a unified paradigm that equips MLLMs with the ability to generate both textual and visual outputs through Visual Reference Tokens (VRTs). By dynamically embedding VRTs into the LLM output space, PaDT ensures semantically coherent and visually grounded predictions, overcoming the inefficiency and misalignment issues of prior codebook-based methods. A light-weight decoder and an effective training strategy are further introduced to enable visual perception and understanding tasks within PaDT. Extensive experiments across detection, segmentation, grounding, and captioning demonstrate state-of-the-art performance, highlighting directly predicting visual tokens as an effective and scalable paradigm toward general-purpose multimodal reasoning systems.

## ACKNOWLEDGEMENT

This work was supported in part by the Guangdong Provincial Key Research and Development Projects (2024B0101040004), the National Research Foundation, Singapore under its National Large Language Models Funding Initiative (AISG Award No: AISG-NMLP-2024-004), the Guangdong Provincial Key Laboratory of Human Digital Twin (2022B1212010004), the NUS Start-up Grant (A-0010106-00-00), and the National Natural Science Foundation of China (62320106007). Any opinions, findings and conclusions or recommendations expressed in this material are those of the author(s) and do not reflect the views of National Research Foundation, Singapore.

ETHICS STATEMENT

We affirm that all authors have read and adhered to the ICLR Code of Ethics. Our research does not involve human subjects, personally identifiable data, or sensitive information. The datasets used are publicly available and cited appropriately. We have considered potential risks, including issues related to fairness, privacy, and security, and have taken steps to mitigate any possible negative impact. No conflicts of interest or external sponsorship have influenced the work. We commit to respecting research integrity and legal compliance throughout the research process.

REPRODUCIBILITY STATEMENT

We are committed to ensuring the reproducibility of our results. The main text provides a detailed description of our proposed method and experimental setup, including all hyperparameters, datasets, and evaluation protocols. Additional results and dataset details are included in the appendix. We also provide the detailed process for constructing the Referring Image Caption dataset in the appendix. We will release all of our implemented code and reproduction instructions to further support the reproducibility of our findings.

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

# A  APPENDIX

## A.1  REFERRING IMAGE CAPTIONING (RIC) DATASET

### A.1.1  DATASET CONSTRUCTION

Image captioning is a fundamental benchmark for evaluating the vision understanding ability of MLLMs. In the conventional setting, given an input image, the model generates a pure textual description that summarizes the main subject and its activity, trained on large-scale image-text pairs. However, such descriptions provide little supervision regarding object-level grounding, making it difficult to assess whether the model accurately captures the spatial locations of entities. To address this limitation, we re-annotate the COCO2017 dataset with more fine-grained annotations and propose our Referring Image Captioning (RIC) dataset. Specifically, we leverage Qwen2.5-VL-72B (prompt provided below) and feed each image together with its box-level annotations, instructing the model to produce enriched captions that explicitly reference the corresponding box IDs. Examples of the resulting training samples are shown in Fig. 6.

---

**Prompt for generating RIC dataset**

**Prompt**: <image>Please describe the image, focusing on the main objects (instances) present. After mentioning an object in the caption, immediately append its instance ID(s) in parentheses, using the format (<box_id: ID/>). You MUST ONLY use the instance IDs provided in the list below. Do NOT invent or make up any IDs. If there are multiple instance IDs for the same object, list all IDs within the same parentheses, separated by commas. For example: 'A man (<box_id: ID/>) is standing by the table (<box_id: ID/>). Two ovens (<box_id: ID/>, <box_id: ID/>) are visible.' Here are the optional instances in the image (id: ID, format: category: name, bbox: [x1, y1, x2, y2]):
- 200887, person, [389, 70, 498, 348]
- {list all annotations within this query image.}

IMPORTANT: Only use the instance IDs listed above in your caption. Do NOT invent or make up any new IDs. Generate a fluent English caption that includes the main objects and their instance IDs as specified.You don't have to include all instances from the list above, as they are optional; only describe the main elements of the image. The caption should be no more than 100 words and should not contain any information irrelevant to the image.

**Qwen2.5-VL-72B**: *(Example)* A bustling kitchen scene features a person (<box_id: 200887/>) working near a sink (<box_id: 2196309/>). The dining table (<box_id: 119568/>) holds various cooking items, including a bowl (<box_id: 713388/>) and a bottle (<box_id: 82445/>). Two ovens (<box_id: 1125079/>, <box_id: 2139366/>) are visible in the background. Another person (<box_id: 1218137/>) appears partially on the left. The table also has a cup (<box_id: 1883614/>) and a knife (<box_id: 693231/>). The kitchen is equipped with hanging pots and a well-used workspace, emphasizing a busy cooking environment.

---

## A.2  TEMPLATE EXAMPLE FOR EACH VISION TASK

Here we present the interaction templates used in our PaDT framework, covering open-vocabulary detection, referring expression comprehension, and referring image captioning tasks. The prompts are designed to be concise and efficient, allowing PaDT to naturally interleave textual tokens and VRT predictions during task training.

---

**Open Vocabulary Detection Task**

**USER**: <image>Please carefully check the image and detect the following objects: [{target list}].
**ASSISTANT**: In this image, there are {number} "{category}" (<VRT_0><VRT_1><VRT_2><VRT_3><VRT_4>, <VRT_5><VRT_6><VRT_7><VRT_8><VRT_9>, ...) and ...

---

**Referring Expression Comprehension / Segmentation Task**

**USER**: <image>Please carefully check the image and detect the object this sentence describes: "{question}".
**ASSISTANT**: The "{question}" refers to <VRT_0><VRT_1><VRT_2><VRT_3><VRT_4> in this image.

---

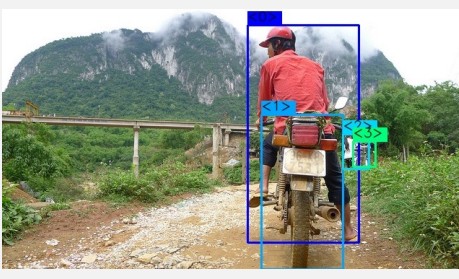

A man (<0>) rides a motorcycle (<1>) on a dirt path. He is wearing a red shirt, black shorts, and a helmet. The motorcycle has a rugged design with a rear storage box. The background features lush greenery, a large mountain partially covered in mist, and a bridge spanning the scene. Another person (<2>) and a bicycle (<3>) are visible in the distance, adding to the rural setting. The atmosphere is serene, with natural elements dominating the landscape.

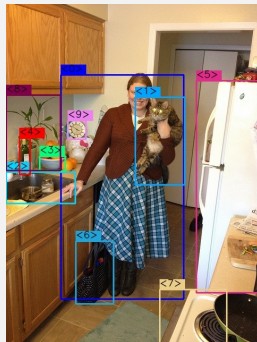

A woman (<0>) stands in a kitchen, holding a cat (<1>). She is near a sink (<2>) with a bowl of oranges (<3>) and a bottle (<4>). A refrigerator (<5>) is visible to her right. A handbag (<6>) rests on the floor by her feet. The kitchen features wooden cabinets and a stove with a pan (<7>). A potted plant (<8>) and a clock (<9>) adorn the counter. The scene is warm and homey.

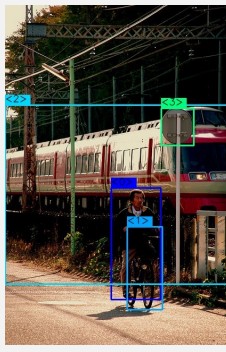

A man (<0>) is riding a bicycle (<1>) on a path near a train (<2>). The train, with a red and white design, is positioned on the tracks behind the cyclist. A stop sign (<3>) is visible to the right of the man, and overhead electrical lines run above the train. The scene is set in a well-lit area with some greenery along the path. The overall atmosphere suggests a calm, possibly suburban environment.

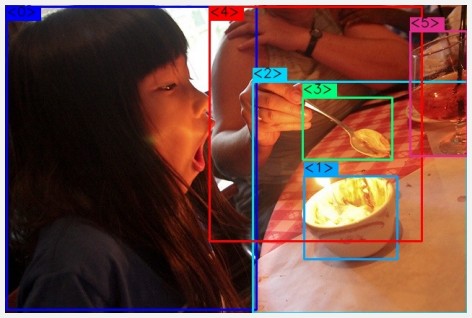

A young girl (<0>) is eagerly eating from a bowl (<1>) on a dining table (<2>). She uses a spoon (<3>) to scoop food, which appears to be melting. The table is covered with a red-checkered tablecloth. In the background, an adult (<4>) sits with arms crossed, and a wine glass (<5>) is partially visible. The scene is illuminated by natural light, creating a warm and casual dining atmosphere.

A woman (<0>) is eating from a spoon (<1>) in a kitchen. She holds a plate of food, including what appears to be broccoli. Behind her, there is an oven (<2>) and a microwave (<3>). A bottle (<4>) is visible on the counter near the oven. The setting is a home kitchen with light-colored cabinets and a tiled backsplash. The woman is wearing glasses and a blue top.

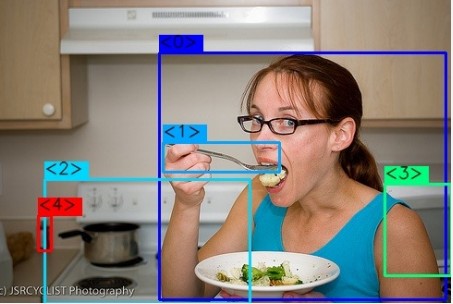

Figure 6: The training samples of RIC dataset. For each image, there are 3-5 captions, in which we ask the MLLMs to refer to the object (via bounding box or visual reference tokens) following each object's subject.

---

**Referring Image Captioning Task**
**USER**: <image>Please describe this image.
**ASSISTANT**: ... {object1} (<VRT_0><VRT_1><VRT_2><VRT_3><VRT_4>) ... {object2} (<VRT_5><VRT_6><VRT_7><VRT_8><VRT_9>) ...

### A.3 PROMPT USED FOR COMPETING METHODS

To guide MLLMs (e.g., Qwen2.5-VL (Bai et al., 2025), InternVL3 (Zhu et al., 2025), and the LLaVA series (Liu et al., 2024c)) in predicting bounding box coordinates in each task, we append a box-specific and format-specific instruction to the task prompt, as detailed below.

---

**Open Vocabulary Detection Task (with box and format instruction)**
USER: <image>Please carefully check the image and detect the following objects: [{target list}]. Output each detected target's bbox coordinates in JSON format. For example, "'json
[{"bbox_2d": [x1, y1, x2, y2], "label": "target name"}]
"'. If no targets are detected in the image, simply respond with None.

---

**Referring Expression Comprehension / Segmentation Task (with format instruction)**
USER: <image>Please carefully check the image and detect the object this sentence describes: "{question}". Output the final answer in JSON format.

---

**Referring Image Captioning Task (with box instruction)**
USER: <image>Please describe this image. You should include the corresponding bounding box of the objects within the sentence. For example, "In this image, a cat ([x1, y1, x2, y2]) is sitting on the wooden table ([x1, y1, x2, y2]), ...".

---

### A.4 THE FORMULA OF THE TASK-SPECIFIC LOSSES ON THE PADT DECODER OUTPUT

Let $\mathcal{B}^{pred} \in \mathbb{R}^{L \times 4}$ denote predicted bounding boxes with ground truth $\mathcal{B}^{gt}$, $\mathcal{M}^{pred} \in \mathbb{R}^{L \times H \times W}$ predicted masks with ground truth $\mathcal{M}^{gt}$, and $\mathcal{S}^{pred} \in \mathbb{R}^{L \times 1}$ predicted confidence scores with ground truth $\mathcal{S}^{gt}$. The $\mathcal{L}_{bbox}$, $\mathcal{L}_{mask}$ and $\mathcal{L}_{score}$ objectives are:

$$\mathcal{L}_{bbox} = \frac{1}{L} \sum_{l}^{L} \mathcal{L}_{iou}(\mathcal{B}_l^{pred}, \mathcal{B}_l^{gt}) + ||\mathcal{B}_l^{pred} - \mathcal{B}_l^{gt}||_1, \tag{8}$$

$$\mathcal{L}_{mask} = \frac{1}{L} \sum_{l}^{L} \mathcal{L}_{dice}(\mathcal{M}_l^{pred}, \mathcal{M}_l^{gt}) + \sum_{l}^{L} \mathcal{L}_{focal}(\mathcal{M}_l^{pred}, \mathcal{M}_l^{gt}), \tag{9}$$

$$\mathcal{L}_{score} = \frac{1}{L} \sum_{l}^{L} ||\mathcal{S}_l^{pred} - \mathcal{S}_l^{gt}||_2^2. \tag{10}$$

### A.5 ADDITIONAL ABLATION STUDY

#### A.5.1 TOKEN ACTIVATION MAP ANALYSIS

We provide additional Token Activation Map (TAM) visualizations, as illustrated in Fig. 7, comparing Qwen2.5-VL and the PaDT Pro 7B model, showing that visual reference tokens establish much stronger associations with target image patches than digit-by-digit coordinate predictions. These results further highlight the robust semantic alignment and precise object localization achieved by visual reference tokens.

#### A.5.2 ABLATION STUDY OF OTHER USED LOSSES

As shown in Table 8, we conduct ablations on the loss components $\mathcal{L}_{bbox}$, $\mathcal{L}_{mask}$, and $\mathcal{L}_{score}$. PaDT achieves the best average performance when all visual task losses are combined. In particular, removing the dynamic embedding module or omitting any individual loss ($\mathcal{L}_{mask}, \mathcal{L}_{bbox}, \mathcal{L}_{score}$) consistently degrades performance on both referring expression comprehension and segmentation. Notably, using all components yields the highest accuracy (93.2% REC and 76.1 mask cIoU) and the strongest multi-task ability, underscoring that each module and loss is essential and complementary for optimal task performance.

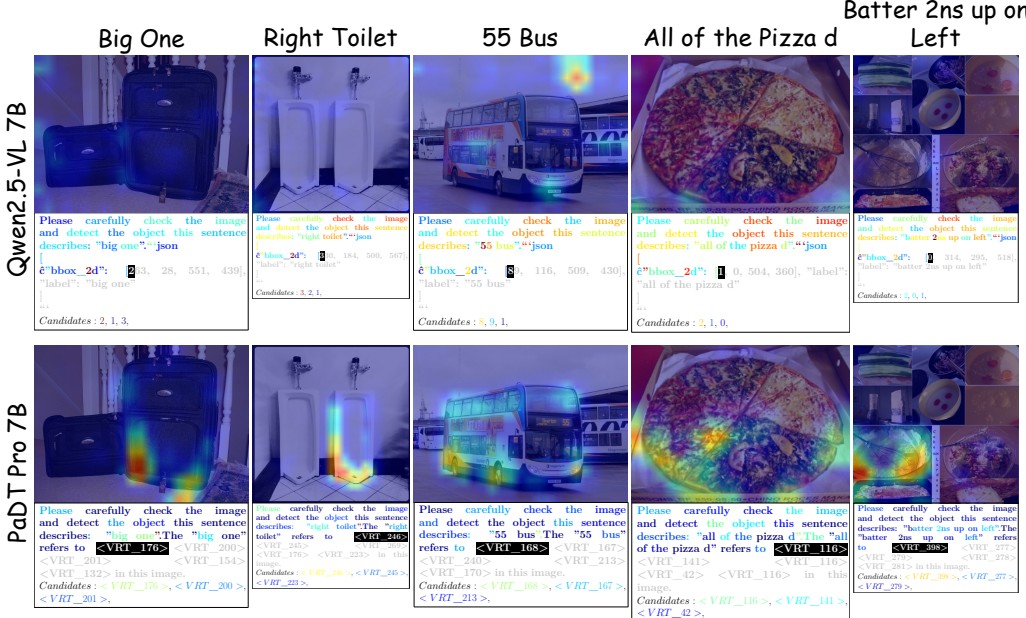

Figure 7: More TAM visualizations of Qwen2.5-VL and our PaDT Pro 7B models.

Table 8: Ablation study of each individual components (with the analysis of additional losses).

| VRT | Dynamic Embedding Module $f_{vp}$ | Visual Task Loss $\mathcal{L}_{mask}$ | $\mathcal{L}_{bbox}$ | $\mathcal{L}_{score}$ | Detection RefCOCO val (REC) | COCO val | Segmentation RefCOCO val (RES) |
|---|---|---|---|---|---|---|---|
| – | – | – | – | – | 88.7 | 17.1 | – |
| ✓ | – | ✓ | ✓ | ✓ | 91.1 | 27.5 | 72.1 |
| ✓ | ✓ | – | ✓ | ✓ | 91.7 | 32.3 | – |
| ✓ | ✓ | ✓ | – | – | – | – | **78.0** |
| ✓ | ✓ | ✓ | ✓ | – | 92.7 | 24.4 | 75.2 |
| ✓ | ✓ | ✓ | ✓ | ✓ | **93.2** | **34.0** | 76.1 |

### A.5.3 ABLATION STUDY OF THE NUMBER OF SELECTED VRTS PER TARGET

We analyze how the number of selected visual patches per target impacts performance. As shown in Table 9, increasing the number of patches from 1 to 5 steadily improves both bounding box accuracy and mask cIoU across all datasets. The best results are obtained with 5 patches per target, while further increasing to 8 patches yields diminishing or even negative returns. This indicates that a moderate number of representative patches provides richer representations, whereas excessive patches introduce noise and redundancy, leading to unstable training of PaDT.

We also investigate the case of using all foreground patches as ground-truth VRTs during training. As shown in Fig. 8, this configuration produces the worst results. Although the number of output VRTs increases, the PaDT decoder exhibits clear performance degradation. We attribute this to the redundancy (that makes the PaDT hard to predict all VRTs at the inference stage) and low resolution of patch-level features: when all foreground patches are used, the decoder is forced to decode trivial and overlapping regions, which prevents it from learning accurate target boundaries and masks, especially when only a limited number of VRTs are predicted at inference. Consequently, selecting a moderate number of informative patches proves more effective than training with all foreground patches.

### A.5.4 ABLATION STUDY OF DIFFERENT SAMPLING STRATEGY

We present a detailed comparison among different sampling strategies, including random sampling (18 patches), using all foreground patches, and border-aware sampling (four tokens from left, top, right and bottom boundaries). The results are summarized in Table 9.

Table 9: Ablation study of the number of selected visual patches per target and different sampling strategy.

| #Patches / Target | | 1 | 3 | **5** | 8 | ALL | Border-aware Sampling |
|---|---|---|---|---|---|---|---|
| RefCOCO val | Bbox Acc@0.5 | 92.4 | 93.2 | **93.2** | 92.6 | 49.1 | 92.1 |
| | Bbox Acc@0.75 | 82.7 | 86.1 | **87.1** | 85.9 | 15.5 | – |
| | Mask cIoU | 67.3 | 75.2 | **76.1** | 75.7 | 19.8 | 70.9 |
| RefCOCO+ val | Bbox Acc@0.5 | 87.5 | 88.1 | **88.5** | 87.5 | – | 86.6 |
| | Bbox Acc@0.75 | 78.8 | 82.1 | **82.8** | 81.7 | – | – |
| | Mask cIoU | 63.7 | 71.4 | **72.7** | 71.6 | – | 66.9 |
| RefCOCOg val | Bbox Acc@0.5 | 88.1 | 88.2 | **88.2** | 86.8 | – | 87.0 |
| | Bbox Acc@0.75 | 78.7 | 80.7 | **81.1** | 79.9 | – | – |
| | Mask cIoU | 62.7 | 69.7 | **70.5** | 70.0 | – | 65.6 |

❌ PaDT trained with all foreground VRTs

✅ PaDT trained with 5 random foreground VRTs

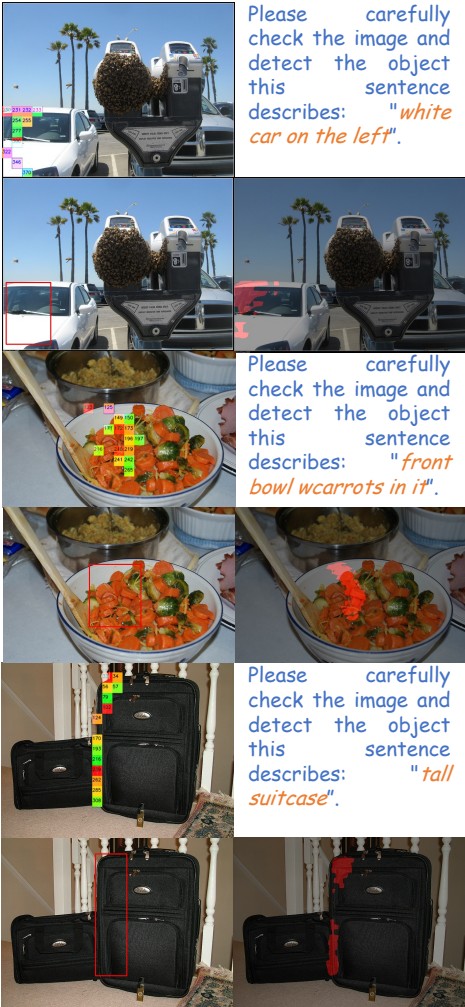
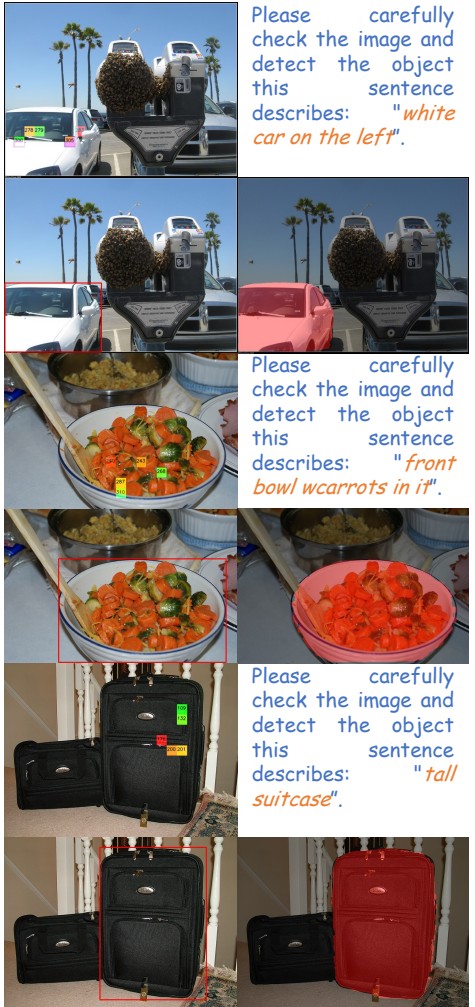

Figure 8: Qualitative analysis between training PaDT with all foreground VRTs and 5 randomly selected foreground VRTs.

We make the following key observations:

- **Using all foreground patches as ground-truth VRTs leads to performance collapse.** When all foreground patches are provided during training, the task decoder tends to overfit to the ground-truth VRTs and relies heavily on the MLLM's predicted VRTs during inference. As the decoder simply learns to produce trivial bounding boxes or masks that cover all foreground areas, it no longer needs to truly understand object boundaries, thus failing to generalize.

- **Random sampling consistently benefits performance.** As the number of randomly sampled patches increases from 1 to 5, the performance consistently improves. The best results are achieved with 5 randomly sampled patches, indicating that this strategy strikes a balance between coverage and model generalization.

- **Boundary-aware sampling underperforms random sampling.** Sampling exclusively from the four boundaries (left, top, right, bottom) yields weaker results. We hypothesize that boundary patches often contain ambiguous semantics, especially when segmentation annotations are unavailable. This increases training difficulty and again makes the task decoder overly dependent on MLLM's predicted boundary VRTs.

## A.6 SCALABILITY TO HIGH-RESOLUTION IMAGES

### A.6.1 COMPATIBILITY WITH HIGH-RESOLUTION IMAGES: YES, FULLY SUPPORTED

PaDT is fully compatible with high-resolution images and supports native resolutions. Our PaDT framework inherits from Qwen2.5-VL, and just like Qwen2.5-VL, it supports image inputs at their original resolution. For instance, in our experiments, we did not perform any resizing operations on training images and we directly use their native resolutions.

- **COCO** dataset: multiple resolutions, such as 640 480, 480 640, 640 573, 500 333, etc. (Tab. 1,2,3).
- **Objects365** dataset: high-resolution images such as 1024 727, 4608 3072, 768 1024, 5152 3864, etc. (Tab. 7).

Table 7 further shows the results of training PaDT on Objects365 (with highly dynamic high-resolution inputs) and transferring to COCO dataset. Both Zero-Shot and Fine-Tuned results (mAP50) outperform Qwen2.5-VL models on the same scale.

### A.6.2 ADDITIONAL COMPUTATION OVERHEAD INTRODUCED BY PaDT: VERY LOW

High-resolution images naturally introduce more visual tokens, leading to increased computation and GPU memory, which is inherent to all MLLM models, including PaDT, Qwen2.5-VL, and InternVL3. Importantly, the additional overhead introduced by PaDT, compared to Qwen2.5-VL, is negligible. We quantify these additional costs as follows:

**1. Number of Visual Tokens / Patches**: For an input image of size $H \times W$, the number of VRTs is: #VRTs = $h \times w$, where $h = round(H/28)$, $w = round(W/28)$. This is identical to the patch extraction process used in Qwen2.5-VL and InternVL3. Thus, PaDT does not introduce new resolution-dependent costs beyond standard visual encoder usage.

**2. Dynamic Embedding Table**:

```
* Qwen2.5-VL-7B Text Embeddings:
  Memory: 152,064 * 3584

* PaDT Dynamic Embedding Table:
  Memory: (152,064 + hw) * 3584
  Additional memory: hw * 3584
  Increasing rate = hw / 152,064
```

For a $1024 \times 1024$ image:

```
h = w = round(1024 / 28) = 37
Extra memory = 37 * 37 * 3584 * 2 Bytes (bfloat16) = 8 MB
Increasing rate = 0.009  (i.e., <1%)
```

**3. Projection Module $f_{vp}$: Negligible Cost**:

```
LayerNorm:
  Memory: 3584 * 2

Two Linear Projections (W_A, W_B):
  Memory: 3584 * 64 * 2
```

This overhead is less than $0.02\%$ of the LLM backbone parameters (3B), thus negligible.

**4. No Extra Overhead in the LLM Forward Pass**: VRTs are treated identically to text tokens during embedding lookup. Once embedded, they are processed by the LLM backbone without any additional computation. No structural modification or auxiliary branch is added to the forward pass.

**5. PaDT Head Overhead**: Let $H \in R^{L \times 3584}$ be the backbone output:

```
* Qwen2.5-VL-7B:
  FLOPs = L * 3584 * 152,064

* PaDT:
  FLOPs = L * 3584 * (152,064 + hw)

Increasing Rate: hw / 152,064 (= 0.009 for a 1024 * 1024 image)
```

Again, the overhead remains less than 1% even for high-resolution images.

**6. Lightweight Decoder Head**: The decoder consists of only three 2-way attention modules, with 95M parameters, significantly smaller than the 37B LLM backbone. Moreover, all VRTs are decoded in a single forward pass, no iterative decoding is required.

**Overall**: PaDT preserves the inference speed and memory footprint of standard multimodal LLMs. The only resolution-dependent cost comes from visual patch extraction, which is inherent to all high-resolution MLLMs.

### A.6.3   PADT IS EVEN MORE EFFICIENT THAN QWEN2.5-VL

Although PaDT introduces negligible overhead, it is more efficient during inference and training. This is because PaDT represents an object with fewer tokens:

```
Qwen2.5-VL:
    [100, 200, 300, 400] -> '[', '1', ..., '0', ']' -> 17 tokens

PaDT:
    <|VRT_0|><|VRT_1|><|VRT_2|><|VRT_3|><|VRT_4|> -> 5 tokens
```

During inference (autoregressive decoding), PaDT saves 12 forward passes per object. During training, it reduces 12 forward tokens per object. These savings greatly outweigh the small memory / computation overhead analyzed above. Therefore, overall, PaDT is both more efficient and more effective than Qwen2.5-VL. More quantitative analysis is shown as Tab. 10, we benchmarked inference on RefCOCO val set (averaged over 100 samples) and scaled the images by 2x (e.g., $448 \times 644 \rightarrow 896 \times 1288$).

### A.7   COMPARISON WITH QWEN2.5-VL USING DIFFERENT POST-TRAINING STRATEGY

We compare PaDT with Qwen2.5-VL under different post-training strategies (i.e., SFT or GRPO) on the task-specific datasets. The results in Table 11 and Table 12 show that: 1. PaDT consistently outperforms post-trained Qwen2.5-VL across both tasks; 2. PaDT achieves superior zero-shot performance; and 3. PaDT demonstrates stronger transferability, as pretraining on Objects365 followed by finetuning on COCO yields better results than training on COCO alone.

Table 10: The quantitative analysis of computation cost and memory allocation for different image resolutions.

| Model | Image Resolution | A Whole Generation Process | Single-Pass Forward | Sequence Length | Peek Memory Allocation |
|---|---|---|---|---|---|
| Qwen2.5-VL (3B) | 1x | 1.127 s | 0.027 s | 42.22 | 8,186 MB |
| PaDT (3B) | 1x | 0.661 s (-0.466 s) | 0.034 s (+0.007 s) | 19.44 (-22.78) | 8,530 MB (+344 MB) |
| Qwen2.5-VL (3B) | 2x | 1.373 s | 0.032 s | 42.96 | 9,470 MB |
| PaDT (3B) | 2x | 0.905 s (-0.468 s) | 0.046 s (+0.014 s) | 19.44 (-23.52) | 9,446 MB (-24 MB) |

Table 11: The results on Referring Expression Comprehension (REC) task.

| Model Name | Setting | RefCOCO val | RefCOCO+ val | RefCOCOg val |
|---|---|---|---|---|
| Qwen2.5-VL | Zero-Shot | 89.1 | 82.4 | 85.2 |
| Qwen2.5-VL | SFT | 88.7 | 82.3 | 86.0 |
| Qwen2.5-VL | GRPO (Shen et al., 2025) | 90.1 | 84.2 | 85.6 |
| PaDT | SFT | 93.2 | 88.5 | 88.2 |
| PaDT-Pro | SFT | **96.0** | **91.8** | **93.6** |

Table 12: The results on Open-Vocabulary Detection (OVD) task.

| Model Name | Setting | mAP@[50:95] |
|---|---|---|
| Qwen2.5-VL | Zero-Shot | 13.7 |
| PaDT | Zero-Shot (Pretrained on Objects365) | **16.9** |
| Qwen2.5-VL | SFT | 17.1 |
| Qwen2.5-VL | GRPO (Shen et al., 2025) | 19.2 |
| PaDT | Task-Specific SFT | 34.0 |
| PaDT | Objects365 $\rightarrow$ COCO | 36.5 |
| PaDT-Pro | SFT | **38.2** |

## A.8 QUALITATIVE EVALUATION

### A.8.1 OPEN VOCABULARY DETECTION ON COCO2017 DATASET

**Comparison with representative MLLMs.** In this section, we present qualitative results for open-vocabulary detection on the COCO2017 dataset, comparing PaDT against representative MLLMs. As shown in Fig. 9, several key observations can be made.

- **Higher recall**. PaDT consistently detects a larger number of objects in the scene, demonstrating stronger recall. This improvement stems from its ability to directly predict visual reference tokens (VRTs) that are anchored to image patches, enabling more reliable coverage of relevant objects.

- **Robustness in cluttered scenes**. Competing MLLMs, which predict serialized bounding box coordinates, often struggle in scenes with many repetitive or similar-looking objects. Their predictions may miss valid instances or collapse onto a few candidates, whereas PaDT maintains distinct references to multiple targets.

- **Avoiding invalid outputs**. Existing MLLMs occasionally fail to produce valid detections, labeled as Error in Fig. 9. In such cases, the models tend to generate repetitive text sequences until reaching the maximum output length, i.e. 2048 tokens. PaDT avoids this failure mode by grounding predictions directly in visual tokens rather than relying solely on text-based serialization.

Overall, these qualitative comparisons reinforce the advantages of PaDT: directly predicting visual tokens not only improves recall but also enhances robustness and stability in open-vocabulary detection.

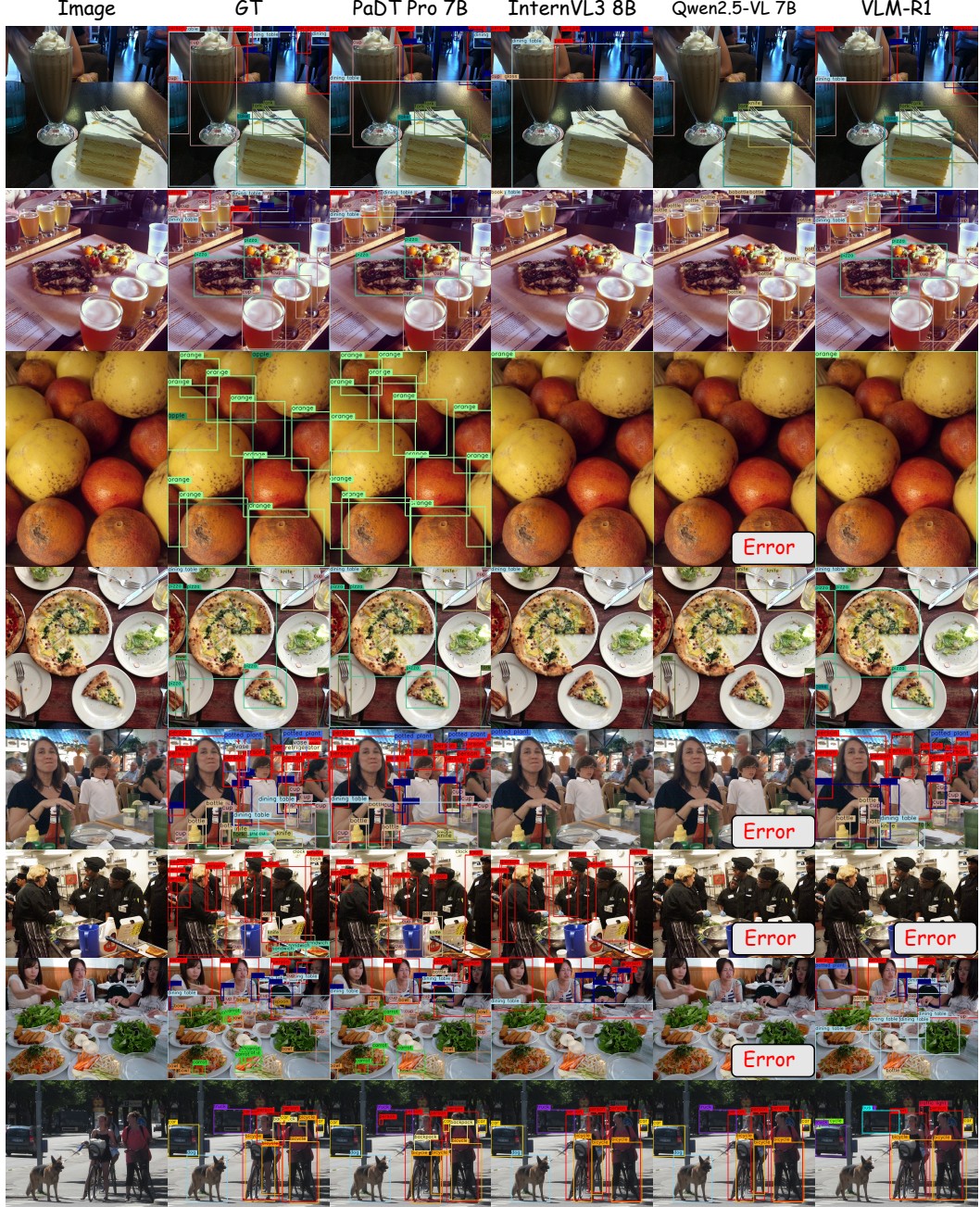

Figure 9: Qualitative comparison on COCO2017 open-vocabulary detection. We compare PaDT with representative MLLMs including InternVL3 and Qwen2.5-VL. Competing models frequently fail to produce valid outputs, leading to Error cases or repetitive text generation. In contrast, PaDT achieves higher recall and correctly identifies multiple objects, even in cluttered scenes with repetitive instances. These results highlight the benefit of directly predicting visual reference tokens over serialized bounding box coordinates.

**Visualization of PaDT results on REC/RES and OVD tasks.** In Fig. 10, we present extensive qualitative examples generated by the proposed PaDT framework. For Referring Expression Comprehension (REC) and Referring Expression Segmentation (RES), PaDT first parses the user query and identifies the corresponding target within the image. As illustrated in the top-left subfigure of each example, PaDT generates five visual reference tokens (VRTs), each directly correlated with specific image patches and thus easily localizable. These VRTs are subsequently passed into the PaDT decoder to produce the corresponding bounding box and segmentation mask. The overall pipeline is simple yet effective. Compared to character-by-character coordinate generation, PaDT requires far fewer tokens (only five VRTs per target) while providing stronger semantic and spatial grounding with respect to the object.

Similar observations are made in the Open-Vocabulary Detection (OVD) task. Unlike REC/RES, OVD requires PaDT to predict multiple targets along with their category labels. In our response template, both categories and VRTs are naturally interleaved within the output sequence, enabling efficient multimodal reasoning. This training strategy strengthens the semantic alignment between text and image patches, thereby improving both precision and recall in detection task.

### A.8.2    REFERRING IMAGE CAPTIONING ON RIC DATASET

**Comparison with representative MLLMs.** In this section, we present qualitative results for open-vocabulary detection on the Referring Image Captioning (RIC) dataset, comparing PaDT with representative MLLMs, including InternVL3 8B and Qwen2.5-VL 7B models. As shown in Fig. 11, PaDT exhibits clear advantages in both bounding box accuracy and object recall. Detailed qualitative comparisons are provided in the figure, further demonstrating the effectiveness of leveraging visual reference tokens as a bridge between high-level text semantics and low-level object localization.

**Visualization of PaDT results on RIC task.** We further present qualitative examples generated by the proposed PaDT framework. As shown in Fig. 12, visual reference tokens are automatically generated alongside the subject, illustrating a natural interleaving between semantic text and image patches. This design further enhances object-level alignment between textual descriptions and visual elements, thereby strengthening the co-reasoning ability across text and image modalities.

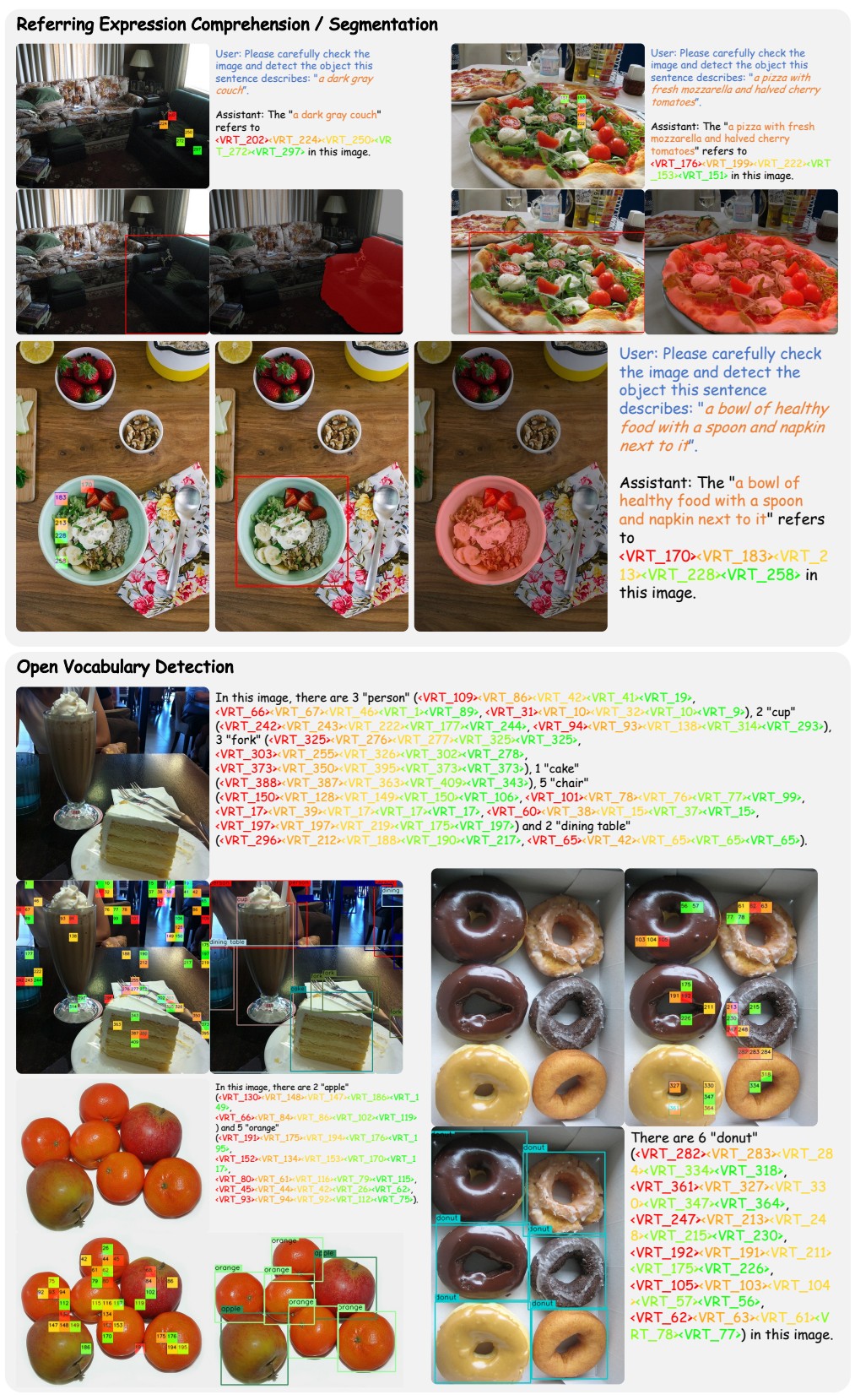

Figure 10: Qualitative visualization of PaDT generated examples on referring expression comprehension/segmentation tasks.

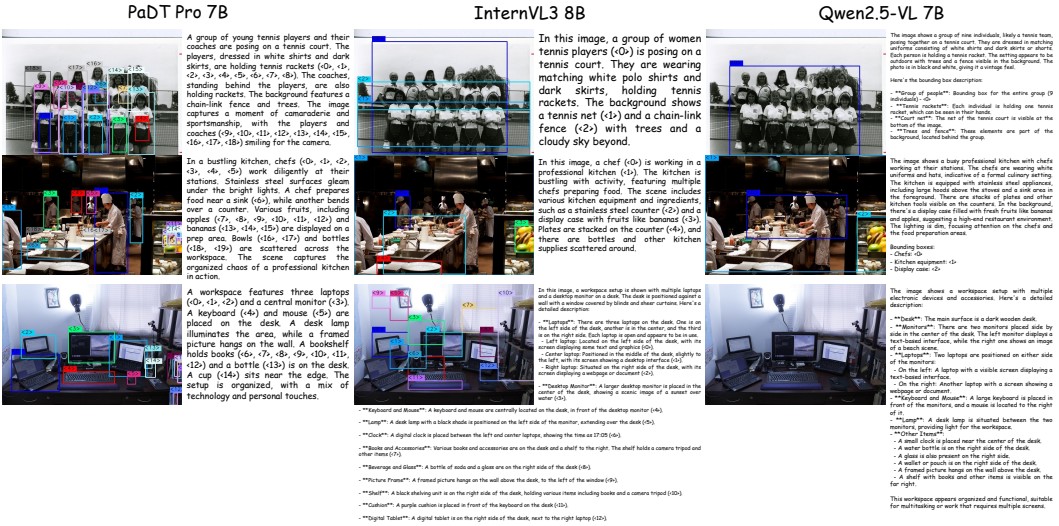

Figure 11: Qualitative comparison on the Referring Image Captioning (RIC) dataset. We compare PaDT with representative MLLMs, including InternVL3 and Qwen2.5-VL. PaDT shows clear advantages in both bounding box accuracy and object recall over competing methods.

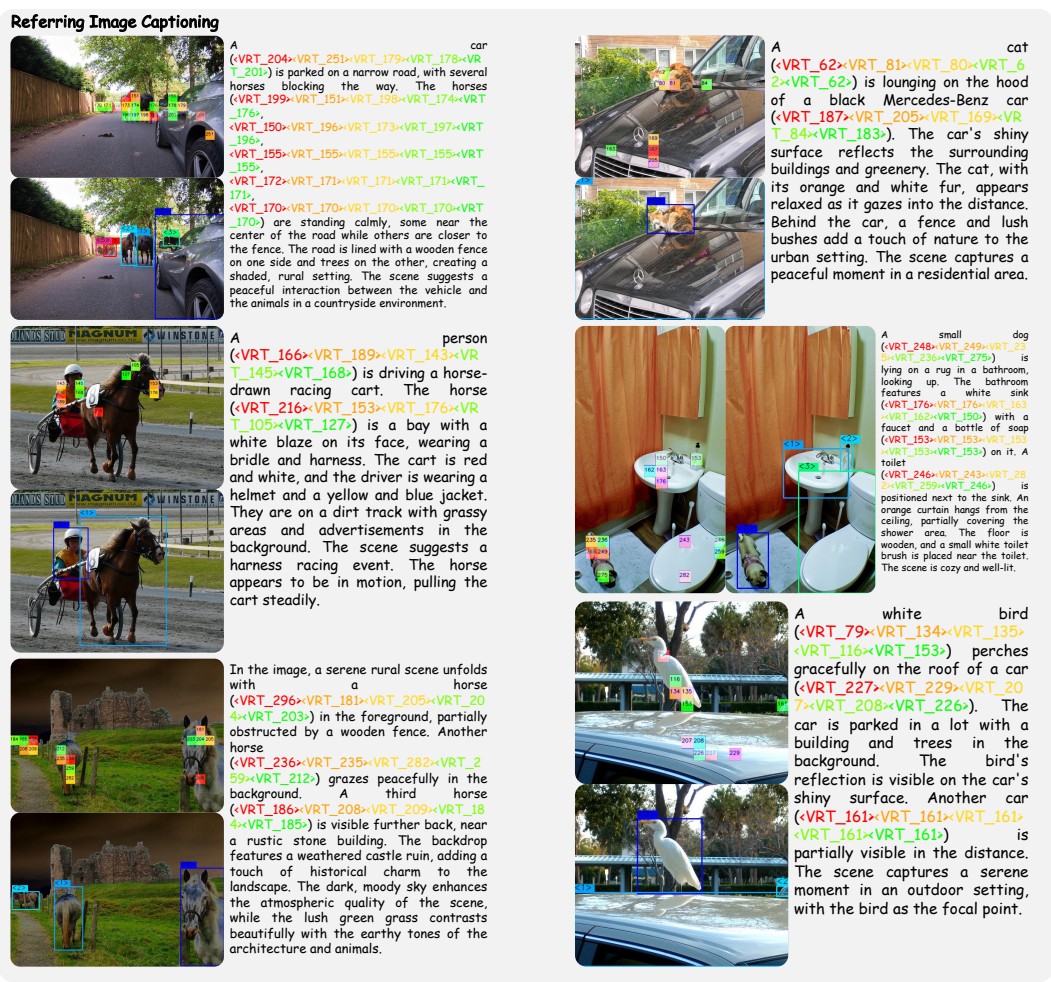

Figure 12: Qualitative visualization of PaDT generated examples on referring image captioning task.

