# OpenReview forum: "Patch-as-Decodable-Token: Towards Unified Multi-Modal Vision Tasks in MLLMs"
_ICLR.cc/2026/Conference — ICLR 2026 Poster_

### Official Review · Reviewer_XCqy · 2025-10-28

**Soundness:** 3
**Presentation:** 3
**Contribution:** 3
**Rating:** 6
**Confidence:** 3

**Summary:**

The paper introduces Patch-as-Decodable Token (PaDT), a unified framework that enables MLLMs to generate both textual and visual outputs. It proposes Visual Reference Tokens, which are learnable tokens derived from image patch embeddings, and they are treated as decodable elements within the LLM’s output sequence. The proposed architecture and training strategy consistently achieve state-of-the-art performance across benchmarks.

**Strengths:**

The proposed framework effectively unifies diverse visual tasks such as detection, segmentation, grounding, and captioning under a single paradigm. The introduced VRT module aligns more naturally with both textual and visual semantics than coordinate-based outputs, reducing inconsistencies and hallucinations. Overall, the approach demonstrates consistent performance improvements over existing models.

**Weaknesses:**

1. The multimodal codebook needs to be adjusted at each forward pass, which may complicate deployment and increase inference latency.
2. The decoder still relies on task-specific components (e.g., bounding box, mask, and score tokens), limiting the framework’s full generality.
3. The evaluation primarily focuses on single-image tasks; it would be valuable to assess the model’s generalization to video or multi-image scenarios.

**Questions:**

1. Table 4 shows that PaDT Pro underperforms PaDT on the referring image captioning task. Could the authors elaborate on the possible reasons for this performance gap?
2. How well does the dynamic vocabulary expansion scale when processing high-resolution images with thousands of patches?
3. What is the memory and computational overhead associated with dynamically updating the embedding table at each forward pass?

---

> ### Author Response · Authors · 2025-11-25
> **Rebuttal Response (1/3)**
>
> Thank you for the thoughtful and careful review, which has greatly helped improve our manuscript. We sincerely appreciate your constructive suggestions regarding the dynamic codebook construction, the generality of the PaDT framework, its potential extension to video/multi-image scenarios, and the analysis of computation/memory overhead. We have carefully addressed all of these concerns with detailed clarifications, analysis, and additional experiments. We hope the following responses will erase your concerns.
>
> ## W1: Dynamic codebook may complicate deployment and increase inference latency.
>
> Thank you for the insightful question. We clarify that the dynamic codebook construction is extremely lightweight and does not introduce practical deployment difficulty or noticeable latency. The process can be viewed simply as a matrix concatenation, as shown in the pseudo-code below:
>
> ```python
> # (Both PaDT and Qwen) Visual patch features extracted from Qwen visual encoder
> image_embeds, high_res_image_embeds, visual_pe = self.visual(pixel_values, grid_thw=image_grid_thw)
>
> # (PaDT) Project visual patch features into Visual Reference Tokens (VRTs)
> image_prototypes = self.vis_norm(image_embeds)
> image_prototypes = image_prototypes + self.vis_proj(image_prototypes)
>
> # (PaDT) Original text codebook
> embed_tokens = self.model.embed_tokens.weight
>
> # (PaDT) Dynamic codebook construction (text codebook + VRTs for the current sample)
> extended_embed_tokens = torch.cat([embed_tokens, image_prototypes], dim=0)
>
> # (Both PaDT and Qwen) Embedding lookup & LLM forward pass (identical to Qwen2.5-VL)
> inputs_embeds = extended_embed_tokens[input_ids]
> ...
>
> # (Both PaDT and Qwen) PaDT head
> logits = hidden_states @ extended_embed_tokens.T
> ```
>
> As shown above, the only extra operations are:
> - A LayerNorm + Linear projection for visual prototype generation.
> - A single matrix concatenation (torch.cat) at each forward pass.
>
> These introduce negligible cost, and do not modify the LLM backbone nor its forward computation. Therefore, PaDT integrates seamlessly into existing MLLM inference pipelines without requiring structural changes.
>
> For further quantitative evidence, we refer the reviewer to the responses to Q2 and Q3, where we provide detailed computation and wall-time benchmark results demonstrating that PaDT introduces <1% computation overhead during each token generation, while reducing total inference time due to fewer generated tokens.
>
>
> ## W2: Task-specific decoder might limit the framework's full generality.
>
> We appreciate the reviewer’s valuable concern. We believe there may be a misunderstanding regarding the core novelty and generality of PaDT. PaDT is built upon the Qwen2.5-VL framework and extends its text-only codebook into a **mixed-modality codebook that includes both text tokens and visual reference tokens (VRTs)**.
>
> Therefore, PaDT naturally inherits all capabilities of Qwen2.5-VL, and can still perform general multimodal tasks without requiring any task-specific decoder, including:
> - image / video VQA
> - image captioning
> - OCR
> - grounding in pure textual format
>
> Our manuscript focuses on tasks such as detection and segmentation, where existing MLLMs (e.g., Qwen2.5-VL) rely on indirect pure-text generation (e.g., producing bounding box coordinates "100 200 300 400"), which is inefficient and misaligned with the nature of visual tasks. This motivates our contribution:
> 1. expanding the LLM codebook with visual tokens,
> 2. and optionally attaching a lightweight decoder to translate VRTs into task-specific outputs (e.g., box/mask prediction).
>
> This enables direct visual prediction, making PaDT efficient and more aligned with the nature of visual tasks.
>
> The task-specific decoder is not required for general multimodal tasks:
>
> - **Without the decoder**: Removing it recovers pure-text generation as in Qwen2.5-VL, or performs cross-modality reasoning but output pure-text.
> - **With the decoder**: Keeping it enables structured visual task outputs, e.g. bounding box detection, segmentation and others (the decoder could be designed for different task).
>
> Hence, the decoder does not constrain the framework’s generality, but rather extends its applicability.
>
> To further demonstrate this, we are conducting additional experiments on general multimodal tasks such as OCR-VQA, TextVQA, and VisualGenome using the LLaVA-665K dataset (following Text4Seg++, which is the journal extension of Text4Seg [ICLR 2025]). These tasks do not require the task-specific decoder and still be compatible by PaDT framework.
> Due to the large training scale, we expect to complete the experiments shortly and will include the results in the camera-ready version.
>
> In summary, PaDT is fully compatible with general multimodal tasks and only employs a lightweight decoder when task-specific structured outputs need. The decoder is a flexible extension, not a limitation, and PaDT retains full generality of existing MLLM frameworks.

---

> ### Author Response · Authors · 2025-11-25
> **Rebuttal Response (2/3)**
>
> ## W3: The experiments on video or multi-image scenarios.
>
> Thank you for this valuable suggestion. We fully agree that extending PaDT to video and multi-image understanding is an exciting and meaningful direction.
>
> Although our current work focuses on single-image tasks, the core design of PaDT, the mixed-modality dynamic codebook and VRT-based representation, is naturally compatible with video scenarios. Specifically:
> - Video frames can be treated as sequential or spatially stacked visual patches, encoded as multi-frame VRTs.
> - The dynamic codebook mechanism allows PaDT to adapt flexibly to varying visual contents across frames, without architectural changes.
> - The decoder can be extended to support temporal reasoning, trajectory prediction, motion-based segmentation, or inter-frame object tracking, making PaDT potentially well-suited for video reasoning tasks.
>
> Indeed, these characteristics suggest that PaDT may be a promising foundation for unified image/video-language modeling. We are actively exploring this direction in the future work, due to limited computational resources and time in the rebuttal stage. We will also release all our PaDT code and implementations, hoping to inspire and support the community in exploring PaDT for video and multi-image understanding.
>
> ## Q1: Analysis of the performance gap in Table 4.
>
> Thank you for this careful observation. In Table 4, we report the results of Referring Image Captioning (RIC), which evaluates two complementary aspects:
>
> - Text Metrics (CIDEr-D, Meteor, ROUGE-L, BLEU-4): measure similarity between generated pure-text captions (removing VRTs) and ground-truth captions.
> - Detection Metrics (Greedy Precision & Recall): measure how accurately the predicted bounding boxes match all possible object boxes in the image.
>
>
> PaDT-Pro includes additional open-vocabulary detection data (e.g., COCO) and referring expression comprehension/segmentation data (e.g. RefCOCO) during training. This improves spatial grounding ability, but it also shifts the optimization focus slightly from linguistic fluency toward object-centric reasoning. As a result, PaDT-Pro tends to generate more object-oriented and fine-grained descriptions, which may differ slightly from ground-truth captions that are more linguistically concise. These differences are penalized by text-only metrics, even though the generated captions remain correct and detailed. **Importantly, the performance gaps are extremely small (e.g., ≤ 0.05 on Meteor), indicating that language generation ability is well preserved.**
>
> On the other hand, **PaDT-Pro exhibits clear advantages in detection metrics**, especially for the 7B model. This aligns with its additional grounding-oriented training data and suggests stronger ability to explicitly localize referenced objects.
>
> We will supplement these detailed analysis in the revision manuscript to assistant readers to better understand the results.
>
> ## Q3: Computational overhead of dynamic codebook expansion and VRT decoding.
>
> Thank you for the question. We confirm that PaDT introduces only negligible computational overhead compared to standard coordinate-based approaches (e.g., Qwen2.5-VL). The increased cost at high resolution mainly stems from the number of visual patches, which is a common phenomenon inherent to all high-resolution MLLMs, rather than specific to PaDT.
>
> Below, we quantify all additional components introduced by PaDT framework:
>
> **(i) Number of Visual Tokens / Patches**
>
> For an input image of size $𝐻 \times 𝑊$, the number of VRTs is:
>
> #VRTs = $h \times w$, where  $h = round(H / 28)$, $w = round(W / 28)$.
>
> This patch extraction is identical to Qwen2.5-VL, thus not a PaDT-specific overhead.
>
> **(ii) Dynamic Embedding Table**
> ```yaml
> * Qwen2.5-VL-7B Text Embeddings:
>   Memory: 152,064 × 3584
>
> * PaDT Dynamic Embedding Table:
>   Memory: (152,064 + hw) × 3584
>   Additional memory: hw × 3584
>   Increasing rate = hw / 152,064
> ```
>
> For a 1024×1024 image:
> ```yaml
> h = w = round(1024 / 28) = 37
> Extra memory = 37 × 37 × 3584 × 2 Bytes (bfloat16) ≈ 8 MB
> Increasing rate = 0.009  (i.e., <1%)
> ```
>
> **(iii) Projection Module $f_{vp}$ (Negligible Cost)**
> ```yaml
> LayerNorm:
>   Memory: 3584 × 2
>
> Two Linear Projections (W_A, W_B):
>   Memory: 3584 × 64 × 2
> ```
> This overhead is <0.02% of the LLM backbone parameters (3B), thus negligible.
>
> **(iv) No Extra Overhead in the LLM Forward Pass**
>
> VRTs are treated identically to text tokens during embedding lookup.
> Once embedded, they are processed by the LLM backbone without any additional computation. No structural modification or auxiliary branch is added to the LLM forward pass.
>
> **(v) PaDT Head Overhead**
>
> Let $H\in R^{L \times 3584}$ be the backbone output:
> ```yaml
> * Qwen2.5-VL-7B:
>   FLOPs = L × 3584 × 152,064
>
> * PaDT:
>   FLOPs = L × 3584 × (152,064 + hw)
>
> Increasing Rate: hw / 152,064 (≈ 0.009 for 1024 × 1024 images)
> ```
>
> Again, the overhead remains <1% even for high-resolution (1024 x 1024) images.

---

> ### Author Response · Authors · 2025-11-25
> **Rebuttal Response (3/3)**
>
> **(vi) Lightweight Decoder Head**
>
> Only three two-way attention modules (~95M parameters), compared to 3–7B in the backbone. All VRTs are decoded within one forward pass, avoiding iterative decoding.
>
> **Overall**, PaDT preserves the inference speed and memory footprint of standard multimodal LLMs. The main resolution-dependent cost comes from visual patch extraction which is inherent to all high-resolution MLLMs, and task decoder which is much smaller than the LLM backbone. We will include these detailed analysis of computation cost in the Appendix for clarity.
>
>
> ## Q2: The inference time comparison between using coordinate-based approaches (e.g. Qwen2.5-VL) and PaDT, and its impact from high-resolution images.
>
> Thank you for raising this important point. As discussed above, the additional computation introduced by PaDT is negligible. In fact, PaDT is faster during inference, primarily because it generates significantly fewer tokens per object, thereby saving a large number of autoregressive generation steps.
>
>
> PaDT reduces generated sequence length significantly:
>
> ```yaml
> Qwen2.5-VL:
>     [100, 200, 300, 400] → '[', '1', ..., '0', ']' → 17 tokens
>
> PaDT:
>     <|VRT_0|><|VRT_1|><|VRT_2|><|VRT_3|><|VRT_4|> → 5 tokens
> ```
>
> Thus, PaDT saves 12 LLM forward steps per object, which brings notable acceleration during inference. The minor overhead introduced by dynamic embedding and VRT decoding is greatly outweighed by the reduction in autoregressive generation.
>
> **Impact of Increasing Image Resolution.**
>
> To further evaluate the inference time of PaDT on high-resolution inputs, we scaled the RefCOCO validation images by 2× (e.g., 448 × 644 → 896 × 1288). We benchmarked inference on the RefCOCO validation set (averaged over 100 samples), and the results are shown below:
>
>
>
> | Model | Image Resolution | A Whole Generation Process (Wall Time) | Single-Pass Forward (Wall Time per Token) | Sequence Length (Tokens) | Peek Memory Allocation |
> |:-:|:-:|:-:|:-:|:-:|:-:|
> |Qwen2.5-VL-3B| 1x | 1.127 s | 0.027 s |  42.22 | 8,186 MB |
> |PaDT-3B| 1x | **0.661 s** (-0.466 s) | 0.034 s (+0.007 s) | 19.44 (-22.78) | 8,530 MB (+344 MB) |
> |Qwen2.5-VL-3B| 2x | 1.373 s | 0.032 s | 42.96 | 9,470 MB |
> |PaDT-3B| 2x | **0.905 s** (-0.468 s) | 0.046 s (+0.014 s) | 19.44 (-23.52) | 9,446 MB (-24 MB) |
>
> Even under higher resolutions, PaDT continues to save computation time in the entire generation process, while maintaining comparable or even slightly lower memory usage compared to Qwen2.5-VL. This demonstrates that PaDT scales well to larger images and maintains efficiency.
>
> We will include the above detailed analysis of computation cost and inference efficiency in the Appendix to further strengthen our manuscript.

---

### Official Review · Reviewer_Cmsv · 2025-11-01

**Soundness:** 3
**Presentation:** 3
**Contribution:** 2
**Rating:** 4
**Confidence:** 3

**Summary:**

This paper introduces Patch-as-Decodable Token (PaDT), a unified paradigm for multimodal large language models (MLLMs) that enables direct generation of both textual and diverse visual outputs. The core innovation is the Visual Reference Token (VRT), derived from image patch embeddings and interleaved with LLM textual tokens. A lightweight decoder transforms these outputs into detection, segmentation, and grounding predictions. The method claims improved localization, semantic alignment, and flexibility over prior approaches that serialize visual outputs as text (e.g., bounding box coordinates). Extensive experiments on detection, segmentation, grounding, and captioning tasks demonstrate state-of-the-art performance, even compared to much larger generalist models.

**Strengths:**

1) The proposed way to bridge textual and visual outputs, allowing MLLMs to generate structured predictions (e.g., bounding boxes, masks) in a unified format. Similar idea was used in Ma et al. 2025, but there is some difference in codebook design. However, I am not sure whether the proposed method is limited to fixed input visual token length, if yes, it will be a limitation compared to Ma et al. 2025.
2) PaDT achieves strong results across multiple tasks and datasets, outperforming larger models (e.g., InternVL3 78B) in detection and segmentation benchmarks. Extensive ablation analysis is also provided.

**Weaknesses:**

1) The paper does not sufficiently address how PaDT handles flexible visual token lengths or variable image resolutions. While adaptive tiling and dynamic codebook expansion are mentioned, it remains unclear whether the method can robustly generalize to images of arbitrary sizes or aspect ratios, especially in real-world scenarios.
2) The comparison with generalist models like QWen and InternVL3 may not be entirely fair. These baselines are designed for broad multimodal tasks, whereas PaDT is specifically trained for detection/segmentation. The paper should clarify whether baselines were fine-tuned for the same tasks or if the comparison is strictly zero-shot. This distinction is crucial for interpreting the reported gains.
3) What is the computational overhead of dynamic codebook expansion and VRT decoding compared to standard coordinate-based approaches? Is inference time affected for high-resolution images?

**Questions:**

Refer to the above part.

---

> ### Author Response · Authors · 2025-11-24
> **Rebuttal Response (1/3)**
>
> Thanks for the thoughtful and detailed review. We sincerely appreciate your constructive suggestions regarding the fair comparison with the compelling methods, as well as the analysis of additional computation cost, inference time, and the impact from high-resolution images. We have carefully addressed all of these concerns with detailed clarification, analysis and experiments, and we hope the following response will erase your concerns.
>
>
> ## Strength 1: Clarification of novelty between PaDT and Ma et al., 2025.
>
> We appreciate the reviewer’s concern regarding the novelty of PaDT compared with ClawMachine (Ma et al., 2025). We believe there might be a misunderstanding about the core difference:
>
> As stated in Sec. 3.2.1, ClawMachine adopts **a fixed VQ-VAE visual codebook**, appended to the text codebook, to enable visual token decoding. However, this design leads to two unresolved issues: (i) **loss of spatial information**, and (ii) **object ambiguity**, due to the lack of direct correspondence between visual tokens and image regions.
>
> In contrast, PaDT does **not rely on a pre-defined codebook**. Instead, it **directly uses visual patch features** (after a linear projection) to construct a sample-specific, mixed-modality, dynamic embedding table that jointly supports text and visual token generation. Crucially, our visual token length is **not fixed**, since the embedding table is dynamically constructed during the forward pass for each sample, as illustrated in Fig. 3. This design **naturally supports varying visual token lengths**, benefiting from the native-resolution input capability of Qwen2.5-VL. Therefore, unlike Ma et al. (2025), our method is not constrained by fixed visual token length.
>
> This dynamic construction (using the original visual patch features) is key to preserving spatial information and avoiding object ambiguity, which is the major methodological distinction and contribution of PaDT.
>
> ## W1: PaDT supports flexible visual token lengths and variable image resolutions.
>
> We appreciate the reviewer’s concern. We clarify that PaDT can robustly handle images with arbitrary resolutions and aspect ratios, and does not require any fixed visual token length.
>
> As stated above and in Sec. 3.2.1, PaDT directly inherits the native-resolution input pipeline of Qwen2.5-VL, thus, no resizing or pre-defined image scaling strategy is applied during training or inference. Instead, PaDT dynamically constructs its mixed-modality embedding table during each forward pass, using the visual patch features extracted from the input image. This design ensures that the number of visual tokens naturally adapts to the input image resolution, and PaDT handles them without architectural modification.
>
> Evidence of robustness to variable resolutions:
>
> - COCO dataset: includes diverse resolutions, e.g., 640×480, 480×640, 640×573, 500×333, etc. (Tables 1–4), and PaDT is trained directly on native images without resizing.
> - Objects365 dataset: contains even more dynamic and higher resolutions, e.g., 1024×727, 4608×3072, 768×1024, 5152×3864, etc. (Table 7).
>
> Importantly, Table 7 shows that PaDT, trained solely on Objects365 (highly dynamic resolutions), can successfully transfer to COCO, achieving superior Zero-Shot and Fine-Tuned results (mAP50) compared with Qwen2.5-VL models of the same scale. This demonstrates that PaDT effectively generalizes to real-world scenarios with arbitrary image sizes and aspect ratios, rather than being limited to a fixed visual token length.

---

> ### Author Response · Authors · 2025-11-24
> **Rebuttal Response (2/3)**
>
> ## W2: Clarification on fairness of comparison with Qwen2.5-VL.
>
> We appreciate the reviewer’s concern regarding the fairness of comparison with generalist models. We agree that it is crucial to distinguish between zero-shot and fine-tuned settings, and we clarify that our comparisons were made under matched training conditions whenever applicable, as summarized below.
>
> We present results under two separate settings:
>
> 1. Zero-shot comparison, where all models are used without any task-specific finetuning (e.g., Table 7).
> 2. Task-specific fine-tuning comparison, where PaDT and the baselines (e.g., Qwen2.5-VL) are fine-tuned using the same optimization strategy and supervision (e.g., SFT or GRPO), as in Tables 1 and 3.
>
> We will restructure our tables in the revision to explicitly separate these two settings for clearer interpretation. Below is an excerpt that demonstrates the fair comparison using the same protocol and model scale:
>
>
> **Referring Expression Comprehension (RefCOCO/+/g) — Fine-tuning setting (SFT/GRPO)**
>
> | Model Name | Setting | RefCOCO val | RefCOCO+ val | RefCOCOg val |
> | :-: | :-: | :-: | :-: | :-: |
> | Qwen2.5-VL | Zero-Shot | 89.1 | 82.4 | 85.2 |
> | Qwen2.5-VL | SFT | 88.7 | 82.3 | 86.0 |
> | Qwen2.5-VL | GRPO (VLM-R1) | 90.1 | 84.2 | 85.6 |
> | PaDT | SFT | 93.2 | 88.5 | 88.2 |
> | PaDT-Pro | SFT | **96.0** | **91.8** | **93.6** |
>
> **Open-Vocabulary Detection (COCO) — Both zero-shot and SFT settings**
>
> | Model Name | Setting | AP@[50:95] |
> | :-: | :-: | :-: |
> | Qwen2.5-VL | Zero-Shot | 13.7 |
> | PaDT | Zero-Shot (Pretrained on Objects365) | **16.9** |
> | -- | -- | -- |
> | Qwen2.5-VL | SFT | 17.1 |
> | Qwen2.5-VL | GRPO (VLM-R1) | 19.2 |
> | PaDT | Task-Specific SFT | 34.0 |
> | PaDT | Objects365 -> COCO | 36.5 |
> | PaDT-Pro | SFT | **38.2** |
>
> These results demonstrate that under both zero-shot and task-specific fine-tuning settings, PaDT consistently outperforms Qwen2.5-VL (PaDT's base model) under the same training protocol and model scale, which we believe establishes a fair and meaningful comparison.
>
>
> ## W3.1: Computational overhead of dynamic codebook expansion and VRT decoding.
>
> Thank you for the question. We confirm that PaDT introduces only negligible computational overhead compared to standard coordinate-based approaches (e.g., Qwen2.5-VL). The increased cost at high resolution mainly stems from the number of visual patches, which is a common phenomenon inherent to all high-resolution MLLMs, rather than specific to PaDT.
>
> Below, we quantify all additional components introduced by PaDT framework:
>
> **(i) Number of Visual Tokens / Patches**
>
> For an input image of size $𝐻 \times 𝑊$, the number of VRTs is:
>
> #VRTs = $h \times w$, where  $h = round(H / 28)$, $w = round(W / 28)$.
>
> This patch extraction is identical to Qwen2.5-VL, thus not a PaDT-specific overhead.
>
> **(ii) Dynamic Embedding Table**
> ```yaml
> * Qwen2.5-VL-7B Text Embeddings:
>   Memory: 152,064 × 3584
>
> * PaDT Dynamic Embedding Table:
>   Memory: (152,064 + hw) × 3584
>   Additional memory: hw × 3584
>   Increasing rate = hw / 152,064
> ```
>
> For a 1024×1024 image:
> ```yaml
> h = w = round(1024 / 28) = 37
> Extra memory = 37 × 37 × 3584 × 2 Bytes (bfloat16) ≈ 8 MB
> Increasing rate = 0.009  (i.e., <1%)
> ```
>
> **(iii) Projection Module $f_{vp}$ (Negligible Cost)**
> ```yaml
> LayerNorm:
>   Memory: 3584 × 2
>
> Two Linear Projections (W_A, W_B):
>   Memory: 3584 × 64 × 2
> ```
> This overhead is <0.02% of the LLM backbone parameters (3B), thus negligible.
>
> **(iv) No Extra Overhead in the LLM Forward Pass**
>
> VRTs are treated identically to text tokens during embedding lookup.
> Once embedded, they are processed by the LLM backbone without any additional computation. No structural modification or auxiliary branch is added to the LLM forward pass.
>
> **(v) PaDT Head Overhead**
>
> Let $H\in R^{L \times 3584}$ be the backbone output:
> ```yaml
> * Qwen2.5-VL-7B:
>   FLOPs = L × 3584 × 152,064
>
> * PaDT:
>   FLOPs = L × 3584 × (152,064 + hw)
>
> Increasing Rate: hw / 152,064 (≈ 0.009 for 1024 × 1024 images)
> ```
>
> Again, the overhead remains <1% even for high-resolution (1024 x 1024) images.
>
> **(vi) Lightweight Decoder Head**
>
> Only three two-way attention modules (~95M parameters), compared to 3–7B in the backbone. All VRTs are decoded within one forward pass, avoiding iterative decoding.
>
> **Overall**, PaDT preserves the inference speed and memory footprint of standard multimodal LLMs. The main resolution-dependent cost comes from visual patch extraction which is inherent to all high-resolution MLLMs, and task decoder which is much smaller than the LLM backbone. We will include these detailed analysis of computation cost in the Appendix for clarity.

---

> ### Author Response · Authors · 2025-11-24
> **Rebuttal Response (3/3)**
>
> ## W3.2: The inference time comparison between using coordinate-based approaches (e.g. Qwen2.5-VL) and PaDT, and its impact from high-resolution images.
>
> Thank you for raising this important point. As discussed above, the additional computation introduced by PaDT is negligible. In fact, PaDT is faster during inference, primarily because it generates significantly fewer tokens per object, thereby saving a large number of autoregressive generation steps.
>
> PaDT reduces generated sequence length significantly:
>
> ```yaml
> Qwen2.5-VL:
>     [100, 200, 300, 400] → '[', '1', ..., '0', ']' → 17 tokens
>
> PaDT:
>     <|VRT_0|><|VRT_1|><|VRT_2|><|VRT_3|><|VRT_4|> → 5 tokens
> ```
>
> Thus, PaDT saves 12 LLM forward steps per object, which brings notable acceleration during inference. The minor overhead introduced by dynamic embedding and VRT decoding is greatly outweighed by the reduction in autoregressive generation.
>
> **Impact of Increasing Image Resolution.**
>
> To further evaluate the inference time of PaDT on high-resolution inputs, we scaled the RefCOCO validation images by 2× (e.g., 448 × 644 → 896 × 1288). We benchmarked inference on the RefCOCO validation set (averaged over 100 samples), and the results are shown below:
>
> | Model | Image Resolution | A Whole Generation Process (Wall Time) | Single-Pass Forward (Wall Time per Token) | Sequence Length (Tokens) | Peek Memory Allocation |
> |:-:|:-:|:-:|:-:|:-:|:-:|
> |Qwen2.5-VL-3B| 1x | 1.127 s | 0.027 s |  42.22 | 8,186 MB |
> |PaDT-3B| 1x | **0.661 s** (-0.466 s) | 0.034 s (+0.007 s) | 19.44 (-22.78) | 8,530 MB (+344 MB) |
> |Qwen2.5-VL-3B| 2x | 1.373 s | 0.032 s | 42.96 | 9,470 MB |
> |PaDT-3B| 2x | **0.905 s** (-0.468 s) | 0.046 s (+0.014 s) | 19.44 (-23.52) | 9,446 MB (-24 MB) |
>
> Even under higher resolutions, PaDT continues to save computation time in the entire generation process, while maintaining comparable or even slightly lower memory usage compared to Qwen2.5-VL. This demonstrates that PaDT scales well to larger images and maintains efficiency.
>
> We will include the above detailed analysis of computation cost and inference efficiency (wall time) in the Appendix to further strengthen our manuscript.

---

> ### Author Response · Authors · 2025-11-27
> **Follow-up on Rebuttal Response**
>
> Dear Reviewer Cmsv,
>
> We would like to once again express our sincere appreciation for your valuable comments, which have significantly contributed to the improvement of our manuscript. In response to your feedback, we have provided a more **detailed clarification of the novelty of our work**, and have included a **comprehensive quantitative analysis of the computational cost and inference overhead** introduced by our PaDT framework. Additionally, we have conducted **experiments with high-resolution images** and have included the results and analysis.
>
> We have also expanded the discussion on the **intuition behind our novel framework** (Reviewer npQb #W4), the **sampling strategy** (Reviewer npQb #W2), and the **detailed analysis supporting our significant results** (Reviewer npQb #W3). Moreover, we have included insights into the **general applicability of the PaDT framework** (Reviewer XCqy #W2), which can be found in our responses to other reviewers.
>
> As the discussion period draws to a close, we would be grateful if you could review our responses. Your constructive feedback has been instrumental in enhancing the quality of our work, and we believe we have addressed all your current concerns thoroughly. If you have no further concerns, we kindly request that you consider updating your rating to 6 or higher.
>
> Thank you once again for your time and the detailed review. We welcome any further questions or requests for clarification.
>
> Yours sincerely,
>
> The Authors

---

### Official Review · Reviewer_npQb · 2025-11-03

**Soundness:** 4
**Presentation:** 3
**Contribution:** 3
**Rating:** 6
**Confidence:** 4

**Summary:**

This paper addresses a core limitation in current Multimodal Large Language Models (MLLMs): their inability to produce fine-grained visual outputs like bounding boxes or segmentation masks directly. Existing methods typically force the model to generate coordinates as a string of text (e.g., [x1, y1, x2, y2]), which is inefficient, prone to formatting errors, and creates a semantic gap between the visual features and the textual output.
To solve this, the authors introduce Patch-as-Decodable Token (PaDT), a new paradigm that allows an MLLM to directly output visual tokens that act as pointers to image patches. The key contributions are:
1. Visual Reference Tokens (VRTs): The core idea is to create special tokens derived from the visual patch embeddings of an input image. These VRTs are dynamically added to the LLM's vocabulary for each specific image.
2. Unified Input-Output: This dynamic vocabulary allows the LLM to seamlessly interleave text and VRTs in its output. For example, to describe a cat, instead of generating coordinates, the model might generate: "Here is the cat <VRT_123><VRT_124><VRT_125>".
3. Lightweight Decoder: A simple, lightweight decoder then takes these few predicted VRTs and translates them into structured visual outputs like bounding boxes or segmentation masks.
4. SoTA Performance: Through a robust training strategy, the authors show that PaDT achieves outstanding results across a range of tasks, including referring expression comprehension, segmentation, and open-vocabulary detection. Notably, their 3B parameter model significantly outperforms much larger models (e.g., 78B InternVL3) that rely on text-based coordinate generation.

**Strengths:**

1. Novel and Elegant Paradigm: The core idea of dynamically creating and predicting patch-level visual tokens (VRTs) is a significant conceptual advance over generating text-based coordinates. It creates a more natural and semantically coherent link between the model's reasoning and the visual content.
2. Empirical Performance: The results are a standout feature of this paper. The PaDT models not only set a new state-of-the-art on multiple challenging benchmarks but do so with remarkable efficiency. The fact that their 3B model outperforms an 78B competitor (Table 1) is a powerful testament to the superiority of their approach. The near-doubling of performance on the COCO open-vocabulary detection task (Table 3) is particularly good.
3. Unified and Flexible Framework: The PaDT method is not a one-trick pony. It provides a single, unified mechanism for handling diverse visual outputs, including bounding boxes and segmentation masks, across different tasks like referring comprehension and open-vocabulary detection. The lightweight decoder design adds to this flexibility.
4. Clarity and High-Quality Presentation: The authors do a great job of explaining a complex idea in a simple and intuitive way. The paper is well-organized, and the figures are highly effective at conveying the core concepts and results.

**Weaknesses:**

Overall, I think the idea is good. I still have some questions regarding the method.

1. Scalability to High-Resolution Images: The number of VRTs in the dynamic vocabulary is directly tied to the number of input image patches. The paper evaluates on standard resolutions, but it's unclear how the method's computational cost (especially memory for the dynamic vocabulary and classifier weights) would scale to very high-resolution images that are divided into thousands of patches. A discussion on this limitation would be welcome.
2. The random sampling of 5 VRTs per object for training is shown to be very effective. What is the intuition behind this random strategy? Have you experimented with more deterministic or "intelligent" sampling methods, for example, using attention maps to select the most salient patches or explicitly selecting patches near object boundaries?
3. The performance on open-vocabulary detection is particularly strong. Do you have a hypothesis for why the VRT approach is so much more effective than text-based coordinates for this specific task?
4. To what extent is the significant performance improvement attributable to simply providing the lightweight decoder with a richer, multi-token representation of the visual target? In other words, is the primary benefit that VRTs provide more detailed visual information to the final prediction head, which prior methods lack, rather than a fundamental improvement in the LLM's spatial reasoning itself?
5. It is better not to only focus on the fine-grained benchmarks, but also try the method on some global semantic tasks, e.g., MMBench, MMVet, to make sure the introduced modules will not hurt on the other important tasks we mostly care about, rather than the specific domains. This makes the whole method more general.

**Questions:**

The proposed method is well-designed and directly targets the clearly articulated limitations of prior work. Please also refer to the weaknesses and hopefully the authors could address them. The paper is technically good. I would like to raise the scores if the authors can address my questions properly.

Besides, some other citations are important, but the authors might be missing:

1. Zhang, Haotian. "Haoxuan You, Philipp Dufter, Bowen Zhang, Chen Chen, Hong-You Chen, Tsu-Jui Fu, William Yang Wang, Shih-Fu Chang, Zhe Gan, et al. Ferret-v2: An improved baseline for referring and grounding with large language models." arXiv preprint arXiv:2404.07973 3 (2024): 21.
2. Lian, Long, et al. "Describe anything: Detailed localized image and video captioning." arXiv preprint arXiv:2504.16072 (2025).

---

> ### Author Response · Authors · 2025-11-25
> **Rebuttal Response (1/4)**
>
> Thanks for the thoughtful and detailed review. We sincerely appreciate your constructive suggestions regarding scalability to high-resolution images, the intuition behind the sampling strategy and the VRT, and the PaDT framework, evaluation on broader multimodal semantic tasks, and valuable related works. We have carefully addressed all of these concerns with detailed clarification, analysis, and additional experiments. We hope that the following response can erase your concerns.
>
> ## W1: Scalability to High-Resolution Images
>
> We appreciate the reviewer for raising this important question regarding the compatibility of the PaDT framework with high-resolution images.
>
> **1. Compatibility with High-Resolution Images: Yes, Fully Supported**
>
> PaDT is fully compatible with high-resolution images and supports native resolutions. Our PaDT framework inherits from Qwen2.5-VL, and just like Qwen2.5-VL, it supports image inputs at their original resolution. For instance, in our experiments, we did not perform any resizing operations on training images and we directly use their native resolutions.
>
> - COCO dataset: multiple resolutions, such as 640 × 480, 480 × 640, 640 × 573, 500 × 333, etc. (Table 1–4).
> - Objects365 dataset: high-resolution images such as 1024 × 727, 4608 × 3072, 768 × 1024, 5152 × 3864, etc. (Table 7).
>
> Table 7 further shows the results of training PaDT on Objects365 (with highly dynamic high-resolution inputs) and transferring to COCO dataset. Both Zero-Shot and Fine-Tuned results (mAP50) outperform Qwen2.5-VL models on the same scale.
>
> **2. Additional Computation Overhead Introduced by PaDT: Very Low**
>
> High-resolution images naturally introduce more visual tokens, leading to increased computation and GPU memory, which is inherent to all MLLM models, including PaDT, Qwen2.5-VL, and InternVL3. Importantly, the additional overhead introduced by PaDT, compared to Qwen2.5-VL, is negligible. We quantify these additional costs as follows:
>
>
> **(i) Number of Visual Tokens / Patches**
>
> For an input image of size $𝐻 \times 𝑊$, the number of VRTs is:
>
> #VRTs = $h \times w$, where  $h = round(H / 28)$, $w = round(W / 28)$.
>
> This is identical to the patch extraction process used in Qwen2.5-VL and InternVL3. Thus, PaDT does not introduce new resolution-dependent costs beyond standard visual encoder usage.
>
> **(ii) Dynamic Embedding Table**
> ```yaml
> * Qwen2.5-VL-7B Text Embeddings:
>   Memory: 152,064 × 3584
>
> * PaDT Dynamic Embedding Table:
>   Memory: (152,064 + hw) × 3584
>   Additional memory: hw × 3584
>   Increasing rate = hw / 152,064
> ```
>
> For a 1024×1024 image:
> ```yaml
> h = w = round(1024 / 28) = 37
> Extra memory = 37 × 37 × 3584 × 2 Bytes (bfloat16) ≈ 8 MB
> Increasing rate = 0.009  (i.e., <1%)
> ```
>
> **(iii) Projection Module $f_{vp}$: Negligible Cost**
> ```yaml
> LayerNorm:
>   Memory: 3584 × 2
>
> Two Linear Projections (W_A, W_B):
>   Memory: 3584 × 64 × 2
> ```
> This overhead is <0.02% of the LLM backbone parameters (3B), thus negligible.
>
> **(iv) No Extra Overhead in the LLM Forward Pass**
>
> VRTs are treated identically to text tokens during embedding lookup.
> Once embedded, they are processed by the LLM backbone without any additional computation. No structural modification or auxiliary branch is added to the forward pass.
>
> **(v) PaDT Head Overhead**
>
> Let $H\in R^{L \times 3584}$ be the backbone output:
>
> ```yaml
> * Qwen2.5-VL-7B:
>   FLOPs = L × 3584 × 152,064
>
> * PaDT:
>   FLOPs = L × 3584 × (152,064 + hw)
>
> Increasing Rate: hw / 152,064 (≈ 0.009 for 1024 × 1024 images)
> ```
>
> Again, the overhead remains <1% even for high-resolution images.
>
> **(vi) Lightweight Decoder Head**
>
> The decoder consists of only three 2-way attention modules,
> with ~95M parameters, significantly smaller than the 3–7B LLM backbone.
> Moreover, all VRTs are decoded in a single forward pass, no iterative decoding is required.
>
>
> **Overall**, PaDT preserves the inference speed and memory footprint of standard multimodal LLMs. The only resolution-dependent cost comes from visual patch extraction, which is inherent to all high-resolution MLLMs. We will include benchmark wall-times and memory usage in the Appendix for clarity.
>
>
> **3. PaDT Is Even More Efficient Than Qwen2.5-VL**
>
> Although PaDT introduces negligible overhead, it is more efficient during inference and training. This is because PaDT represents an object with fewer tokens:
>
> ```yaml
> Qwen2.5-VL:
>     [100, 200, 300, 400] → '[', '1', ..., '0', ']' → 17 tokens
>
> PaDT:
>     <|VRT_0|><|VRT_1|><|VRT_2|><|VRT_3|><|VRT_4|> → 5 tokens
> ```

---

> ### Author Response · Authors · 2025-11-25
> **Rebuttal Response (2/4)**
>
> During inference (autoregressive decoding), PaDT saves 12 forward passes per object. During training, it reduces 12 forward tokens per object. These savings greatly outweigh the small memory/computation overhead analyzed above. Therefore, overall, PaDT is both more efficient and more effective than Qwen2.5-VL. More quantitative analysis is shown below, we benchmarked inference on RefCOCO val set (averaged over 100 samples) and scaled the images by 2x (e.g., 448 x 644 → 896 × 1288).
>
> | Model | Image Resolution | A Whole Generation Process (Wall Time) | Single-Pass Forward (Wall Time per Token) | Sequence Length (Tokens) | Peek Memory Allocation |
> |:-:|:-:|:-:|:-:|:-:|:-:|
> |Qwen2.5-VL-3B| 1x | 1.127 s | 0.027 s |  42.22 | 8,186 MB |
> |PaDT-3B| 1x | **0.661 s** (-0.466 s) | 0.034 s (+0.007 s) | 19.44 (-22.78) | 8,530 MB (+344 MB) |
> |Qwen2.5-VL-3B| 2x | 1.373 s | 0.032 s | 42.96 | 9,470 MB |
> |PaDT-3B| 2x | **0.905 s** (-0.468 s) | 0.046 s (+0.014 s) | 19.44 (-23.52) | 9,446 MB (-24 MB) |
>
>
> ## W2.1: The intuition behind the random strategy.
>
> Thank you for the valuable suggestion regarding the sampling strategy of Visual Reference Tokens (VRTs). Our motivation for adopting a random sampling strategy is to enhance the prediction diversity and encourage the MLLM to learn a holistic representation of each target object. By randomly selecting a fixed number of VRTs per object, the model is encouraged to perceive the entire foreground region, rather than overfitting to a specific area or pattern.
>
> We have also explored deterministic strategies, such as sampling VRTs along the object boundary or at predefined positions within the bounding box. However, these strategies consistently underperformed compared to random sampling. Our hypothesis is twofold. First, random sampling exposes the MLLM to diverse spatial positions of the object during training, which enhances its overall spatial awareness and leads to richer visual representations. This, in turn, benefits the lightweight decoder and results in more precise localization and segmentation. Second, when segmentation masks are not provided (e.g., detection-only datasets, Objects365), boundary-based sampling is prone to selecting patches that actually correspond to the background rather than the object foreground at the boundary of the box. These ambiguous or mislabeled regions significantly hinder training stability and convergence.
>
> Therefore, we believe that random sampling strikes a good balance between coverage and variability, preventing the model from overfitting to specific sampling locations and promoting better generalization across the full object extent.
>
>
>
>
> ## W2.2: Comparison with More Deterministic or “Intelligent” Sampling Strategies.
>
> To further address the reviewer’s suggestion, we present a detailed comparison among different sampling strategies, including random sampling (1–8 patches), using all foreground patches, and border-aware sampling. The results are summarized in the table below.
>
> We make the following key observations:
> 1. **Using all foreground patches as ground-truth VRTs leads to performance collapse.** When all foreground patches are provided during training, the task decoder tends to overfit to the ground-truth VRTs and relies heavily on the MLLM's predicted VRTs during inference. As the decoder simply learns to produce trivial bounding boxes or masks that cover all foreground areas, it no longer needs to truly understand object boundaries, thus failing to generalize.
> 2. **Random sampling consistently benefits performance.** As the number of randomly sampled patches increases from 1 to 5, the performance consistently improves. The best results are achieved with 5 randomly sampled patches, indicating that this strategy strikes a balance between coverage and model generalization.
> 3. **Boundary-aware sampling underperforms random sampling.** Sampling exclusively from the four boundaries (left, top, right, bottom) yields weaker results. We hypothesize that boundary patches often contain ambiguous semantics, especially when segmentation annotations are unavailable. This increases training difficulty and again makes the task decoder overly dependent on MLLM's predicted boundary VRTs.
>
> | | 1 | 3 | 5 | 8 | ALL | Boundary-aware Sampling (4 boundaries: left, top, right, bottom) |
> | - | :-: | :-: | :-: | :-: | :-: | :-: |
> |RefCOCO  val Box Acc | 92.4 | 93.2 | **93.2** | 92.6 | 49.1 | 92.1 |
> |RefCOCO+ val Box Acc | 87.5 | 88.1 | **88.5** | 87.5 |  --  | 86.6 |
> |RefCOCOg val Box Acc | 88.1 | 88.2 | **88.2** | 86.8 |  --  | 87.0 |
> |RefCOCO  val Mask cIoU | 67.3 | 75.2 | **76.1** | 75.7 | 19.8 | 70.9 |
> |RefCOCO+ val Mask cIoU | 63.7 | 71.4 | **72.7** | 71.6 |  --  | 66.9 |
> |RefCOCOg val Mask cIoU | 62.7 | 69.7 | **70.5** | 70.0 |  --  | 65.6 |

---

> ### Author Response · Authors · 2025-11-25
> **Rebuttal Response (3/4)**
>
> ## W3: Regarding the stronger performance of VRTs in open-vocabulary detection
>
> Thank you for highlighting this aspect. We believe the remarkable improvement of the VRT approach on open-vocabulary detection stems from its fundamental change in how the MLLM interacts with visual information during prediction.
>
>
> Conventional methods, e.g. Qwen2.5-VL and InternVL3, represent location outputs as text-based coordinates (e.g., "[x1,y1,x2,y2]"), treating the task as a text generation problem. This format introduces a semantic gap, as these character tokens (i.e. `400->4,0,0`) have weak semantic relevance to visual objects, and the MLLM's reasoning takes place entirely in the textual space (as shown in Fig 2(b) in the manuscript), which is not inherently suited for precise visual localization and object recall. As a result, the model’s spatial reasoning and localization accuracy are limited, especially in open-vocabulary scenarios where the number of objects are numerous and object categories are not constrained or predefined.
>
> In contrast, our PaDT enables the MLLM to output visual reference tokens (VRTs) that directly reference specific image regions. This tightly couples the localization process with the model’s visual understanding, as the MLLM can “point” to image patches rather than describing boundaries via text. During training and inference, this anchors the reasoning process in the visual feature space, which greatly enhances the model’s ability to locate arbitrary objects and transfer knowledge to unseen categories.
>
> For other tasks, i.e. REC, where the overall accuracy is already high and only a single object is involved, the difference between VRTs and text-based formats may be less significant. However, for open-vocabulary detection, which places a strong emphasis on the semantic retrieval ability, generalization and precise localiation of MLLM, the VRT paradigm provides a distinct and significant advantage. We also believe this fundamental shift in paradigm lays a promising foundation for further improvements in various visual reasoning tasks.
>
>
> ## W4: Regarding the source of performance gains: richer multi-token representation vs. enhanced spatial reasoning
>
> Thank you for this insightful question. We are glad to clarify why the benefits of PaDT extend beyond merely providing a richer multi-token representation to the lightweight decoder. As shown in the table above, our PaDT achieves outstanding performance although using just one VRT.
>
> Firstly, the motivation behind PaDT stems from a key limitation in current MLLMs (e.g., Qwen2.5-VL, InternVL3): object localization is performed by generating coordinate numbers character-by-character. As shown in Fig. 2(b) of the manuscript, there exists no direct semantic correspondence between these numeric tokens and the visual object they aim to localize. Thus, the model’s reasoning is fundamentally carried out in textual space, where visual content is only weakly associated with the generated numbers.
>
> One might ask: if this approach is flawed, why do MLLMs still perform well on RefCOCO? We believe this is largely due to two factors:
> 1. Each REC sample contains only one object, keeping the reasoning task relatively simple.
> 2. MLLMs implicitly store localization hints in intermediate textual tokens (e.g., “check”, “coordinate”), which function similarly to [CLS] tokens in ViT. The final coordinate digits are then decoded by attending to these intermediate words rather than the actual visual patches, as observed in Fig. 2(b), where the token `4` attends to `check`, not to the corresponding visual patches.
>
> ```
> > MLLM on RefCOCO:
>     Text description --[semantic association]--> Image patches --[collected by]--> Intermediate textual tokens (check, coordinate) --[decoding]--> Coordinate
>
> > MLLM on COCO:
>     Category Text (e.g. dog) --[semantic association]--> {dog1 patches, dog2 patches, dog3 patches, ...} --[collected by]--> Intermediate textual tokens (!ambiguous) --[decoding]--> {coordinate1, coodinate2, coordinate1, coordinate2, ...} ❌
>
>     * Duplicate generation observed in ChatRex (IDEA, 2024)
>
> > PaDT on COCO:
>     Category Text (e.g. dog) --[semantic association]--> {dog1 VRTs, dog2 VRTs, dog3 VRTs, ...} --[task decoder]--> {coordinate1, coordinate2, coordinate3, ...} ✅
> ```
>
> However, this workaround breaks down in more complex settings, especially: **dense predictions / multi-object scenes, open-vocabulary detection and precise localization tasks**, since MLLMs decoding coorindates neeed intermediate tokens but the targets are numerous. Moreover, character-level coordinate generation disrupts spatial continuity, making it incompatible with standard optimization objectives such as IoU / Dice loss.

---

> ### Author Response · Authors · 2025-11-25
> **Rebuttal Response (4/4)**
>
> **Core contribution of PaDT:** To address this fundamental issue, our goal is to find a representation that remains semantically compatible with text, while (i) carrying explicit spatial information, (ii) being decodable into structured outputs (e.g., box / mask), (iii) and allowing the LLM’s reasoning using them and text tokens in the semantic space through attention machanisms.
>
> This led to the introduction of Visual Reference Tokens (VRTs), direct projections of visual patch features. **VRTs make visual patches direct decodable.** Importantly:
> - VRTs participate in the MLLM’s attention and reasoning, together with text tokens, in the same semantic space.
> - They serve as both semantic carriers and localization anchors, allowing the LLM to “think visually”, not merely output text-formatted coordinates.
> - The lightweight decoder simply translates VRTs into task-specific outputs, analogous to decoding from visual proposals in detectors such as DETR.
>
> Thus, the improvement is not solely from a richer multi-token input, but from enabling a fundamental shift in reasoning modality, i.e. from text-space reasoning to mix-modality reasoning.
>
>
> ## W5: Evaluation on broader multimodal semantic tasks (e.g., MMBench, MMVet).
>
> Thank you for this valuable suggestion. We agree that evaluating PaDT on broader multimodal benchmarks, such as MMBench and MMVet, is important for demonstrating its generality beyond fine-grained localization tasks.
>
> Our current manuscript primarily focuses on introducing the patch-as-decodable-token (PaDT) paradigm, which addresses the limitations of coordinate-based prediction in existing MLLMs. To clearly showcase the effectiveness and efficiency of: (i) the mixed-modality dynamic codebook (text + visual tokens), and (ii) the flexible mixed-modality output format. We concentrate our experiments on OVD, REC, and RES, where the proposed paradigm provides the most direct benefits and enables fair comparison with existing baselines (as discussed in response to Reviewer Cmsv #W2).
>
> Importantly, PaDT retains all abilities of the base MLLM backbone (e.g., Qwen2.5-VL), and can still perform general pure-text reasoning and text-based output generation. However, these general tasks make it difficult to directly observe the advantages of VRT, since the training data rarely involves scenarios where VRT is actually needed. This aspect is detailed in our response to Reviewer XCqy #W2, indicating that the introduced modules do not compromise general reasoning capabilities.
>
> Due to computational and time constraints, we were unable to scale PaDT training to the level required to compete with large-scale enterprise models (e.g., Qwen2.5-VL, InternVL3) on general benchmarks.
>
> To address this concern, we have already initiated an experiment where PaDT is trained jointly on COCO (OVD & REC/RES tasks) and LLaVA-665K (general VQA, OCR, and reasoning tasks). This experiment is currently running, and we plan to include the results as soon as the experiment is finished before the camera-ready stage, to further demonstrate the generality of PaDT across both fine-grained and globally semantic tasks.
>
> We appreciate this suggestion and will continue to explore this direction actively.
>
>
> ## Question: Regarding missing related works.
>
> Thank you for pointing out these valuable references. We have carefully read the suggested works and agree that they are relevant to our research. We will include them in the revised manuscript and update both the related work section and the corresponding comparison tables accordingly.
>
> Specifically:
> - Ferret-v2 (Zhang et al., 2024) will be added to Table 1, as it provides an improved baseline for referring and grounding with MLLMs.
> - DAM (Lian et al., 2025) will be incorporated into the discussion of referring image captioning task. DAM is designed for detailed localized captioning which is related to the target of referring image captioning task, so we will include the discussion with DAM in this related part.
>
> We sincerely appreciate your careful suggestions and will make these additions in the revision.
>
> [1] Zhang, Haotian. "Haoxuan You, Philipp Dufter, Bowen Zhang, Chen Chen, Hong-You Chen, Tsu-Jui Fu, William Yang Wang, Shih-Fu Chang, Zhe Gan, et al. Ferret-v2: An improved baseline for referring and grounding with large language models." arXiv preprint arXiv:2404.07973 3 (2024): 21.
> [2] Lian, Long, et al. "Describe anything: Detailed localized image and video captioning." arXiv preprint arXiv:2504.16072 (2025).

---

### Author Response · Authors · 2025-12-03
**Conclusion of Discussion Stage**

Dear Reviewers and Area Chairs,

We sincerely appreciate the constructive and professional feedback provided by the reviewers, which has significantly improved our manuscript in many aspects. All revisions are highlighted in $\color{blue}{\text{blue}}$ in the updated paper. Below, we summarize the major changes:

1. We have added **the key related work** (i.e., Ferret-v2 and DAM, as suggested by Reviewer npQb) to the **Related Work** section. In addition, we included **the comparable results of Ferret and Ferret-v2 in Table 1** (Referring Expression Comprehension task) to enhance the completeness of our comparison.

2. We appreciate that all three reviewers emphasized the importance of evaluating **the scalability of PaDT under high-resolution images**. In response, we conducted a quantitative analysis of the additional computation cost and wall-clock time introduced by PaDT on top of the Qwen2.5-VL framework. We provide detailed comparisons of computation cost and wall-clock time under two image resolutions for both PaDT and Qwen2.5-VL. These results and clarifications are added in **Section A.6** (including A.6.1–A.6.3) of the Appendix.

3. We supplemented detailed results for **different VRT sampling strategies**, including randomly selecting 1 to all foreground patches and selecting VRTs at the four spatial boundaries. These results are presented in **Section A.5.4** of the Appendix.

4. Following Reviewer Cmsv’s suggestion, we **restructured the fairness comparison between PaDT and Qwen2.5-VL** by ensuring that both methods are evaluated under consistent settings (i.e., same post-training strategy or zero-shot). We provide the results in **Section A.7** of the Appendix, where we compare PaDT and Qwen2.5-VL using SFT, GRPO, and zero-shot settings.

Additionally, as kindly suggested by Reviewer npQb, our general model PaDT-General, which extends the PaDT framework to broader tasks including pure-text VQA, general semantic understanding, and multi-round conversation, is ready for release soon (The experiments are finished in a few days). We will include these results, illustrative examples, and analyses in the camera-ready version, and will release all PaDT checkpoints and reproducibly codes to further support the community.

With the responses in the discussion stages and updates in the revised manuscript, we believe that we have fully addressed all concerns raised by the reviewers. We would once again like to express our sincere gratitude for the constructive comments. We welcome any further suggestions or recommendations that may help us strengthen this work.

Best regards,

The Authors

---

### Meta-Review · Area_Chair_deyn · 2026-01-08

**Summary:**

This paper proposes Patch-as-Decodable Token (PaDT), a “paradigm” to unify textual and visual outputs. The main idea is to generate Visual Reference Tokens (VRTs) via “Dynamic Embedding Module” that are then decoded to visual outputs via “Light-Weight PaDT Decoder”. The paper receives 6, 6, 4 ratings: Reviewer npQb (6), Reviewer Cmsv (4), Reviewer XCqy (6).

The concerns can be divided into 2 groups.

1. The idea and its implementation

1.1 Significance of the idea: Reviewer Cmsv observes similarity to Claw Machine (Ma et al. 2025).

1.2 Handling of variable visual token lengths. Reviewer Cmsv.

1.3 Computational Overhead and High-resolution images. Reviewer npQb, Reviewer Cmsv

1.4 Inherent limitation of the approach: Reviewer XCqy has a concern over the complexity of the dynamic codebook deployment and the reliance on task-specific (object detection, segmentation, etc.) decoder components

2. Experiments

2.1 Fairness of comparison to generalist models: Reviewer Cmsv requests clearer zero-shot and fine-tuned comparison.

2.2 Benchmark scope: Reviewer XCqy would like to see results beyond a single image output. Reviewer npQb would like to see performance on broader benchmarks.

**Reviewer Concerns:**

Overall, the concerns regarding the efficiency and complexity (1.2, 1.3, 1.4) of the approach are robustly resolved. 2.1 is also resolved.

For 1.1, novelty with respect to Claw Machine exists but is likely not ground-breaking. 2.2 is not resolved with the results on generalist multimodal benchmarks on the way and the results on multi-image output left as future work. Overall, the paper still has strong experimental results.

**Reviewer Scores:**

Reviewer npQb (6+): Likely to increase

Reviewer Cmsv (4+): Likely to increase

Reviewer XCqy (6?): Keep or increase

---

### Decision · Program_Chairs · 2026-01-26

Accept (Poster)